# Interaction of tankyrase and peroxiredoxin II is indispensable for the survival of colorectal cancer cells

Dong Hoon Kang[1,2], Doo Jae Lee[1,2], Sunmi Lee[1], So-Young Lee[1], Yukyung Jun[1], Yerin Kim[1], Youngeun Kim[3], Ju-Seog Lee[4], Dae-Kee Lee[1], Sanghyuk Lee[1,2], Eek-Hoon Jho[3], Dae-Yeul Yu[5] & Sang Won Kang[1,2]

Mammalian 2-Cys peroxiredoxin (Prx) enzymes are overexpressed in most cancer tissues, but their specific signaling role in cancer progression is poorly understood. Here we demonstrate that Prx type II (PrxII) plays a tumor-promoting role in colorectal cancer by interacting with a poly(ADP-ribose) polymerase (PARP) tankyrase. *PrxII* deletion in mice with inactivating mutation of adenomatous polyposis coli (*APC*) gene reduces intestinal adenomatous polyposis via Axin/β-catenin axis and thereby promotes survival. In human colorectal cancer cells with *APC* mutations, PrxII depletion consistently reduces the β-catenin levels and the expression of β-catenin target genes. Essentially, PrxII depletion hampers the PARP-dependent Axin1 degradation through tankyrase inactivation. Direct binding of PrxII to tankyrase ARC4/5 domains seems to be crucial for protecting tankyrase from oxidative inactivation. Furthermore, a chemical compound targeting PrxII inhibits the expansion of APC-mutant colorectal cancer cells in vitro and in vivo tumor xenografts. Collectively, this study reveals a redox mechanism for regulating tankyrase activity and implicates PrxII as a targetable antioxidant enzyme in *APC*-mutation-positive colorectal cancer.

[1] Department of Life Science, Ewha Womans University, Seoul 120-750, Korea. [2] Research Center for Cell Homeostasis, Ewha Womans University, Seoul 120-750, Korea. [3] Department of Life Science, University of Seoul, Seoul 130-743, Korea. [4] Department of Systems Biology, Division of Cancer Medicine, UT MD Anderson Cancer Center, Houston, TX 77054, USA. [5] Disease Model Research Laboratory, Aging Research Center, KRIBB, Daejeon 305-333, Korea. Dong Hoon Kang and Doo Jae Lee contributed equally to this work.   Correspondence and requests for materials should be addressed to S.W.K. (email: kangsw@ewha.ac.kr)

More than 50% of patients with colorectal cancer (CRC) have inactivating mutations in the adenomatous polyposis coli (APC) tumor suppressor gene[1]. Since APC is a key scaffold protein in the β-catenin destruction complex, the APC mutations induce the Wnt-independent accumulation of transcriptionally active β-catenins and thus initiate intestinal tumorigenesis[2, 3]. Axis inhibition protein 1 (Axin1) tumor suppressor is another scaffold protein in the β-catenin destruction complex, but endogenous Axin1 proteins are tightly controlled by tankyrase-dependent degradation in CRC cells[4]. Tankyrases (TNKS1/2; also known as PARP5/6 and ARTD5/6) are very distinct poly(ADP-ribose) polymerase (PARP) family enzymes that contain ankyrin repeat regions, involved in the substrate binding, and a oligomerization domain called a sterile alpha motif[5]. Since TNKS regulates telomere length in addition to Wnt signaling, it has emerged as a key therapeutic target for treating CRC. However, the molecular mechanisms regulating the TNKS activity in CRC are largely unknown.

Recently, numerous studies have indicated that intestinal tumorigenesis initiated by APC mutations is promoted by the acquired or inherited mutation in the DNA glycosylase enzymes essential for base excision repair of oxidative DNA damage[6], which suggests that elevation of reactive oxygen species (ROS) levels is certainly involved in the APC mutation-driven intestinal tumorigenesis. Nonetheless, treatment of CRC targeting endogenous redox systems has not been attempted to date. As the $H_2O_2$ of ROS converts to the hydroxyl radical capable of causing DNA damages, cancer cells inherently harbor a high risk of genetic mutations[7]. Hence, cancer cells survive intrinsic ROS cytotoxicity by overexpressing antioxidant enzymes, such as peroxiredoxin (Prx, gene loci Prdx)[8, 9]. Mammalian 2-Cys Prx enzymes are actually the most efficient peroxidases that catalyze the reduction of $H_2O_2$ to water in the presence of NADPH by coupling with the thioredoxin/thioredoxin reductase system[10]. Paradoxically, $H_2O_2$ also functions as a potential second messenger in the proliferating cancer cells via reversible oxidation of the signaling proteins, including protein kinases or protein tyrosine phosphatases[11–14]. In the signaling context, we have demonstrated that a cytosolic 2-Cys Prx isoform, PrxII, regulates the localized $H_2O_2$ in platelet-derived growth factor and vascular endothelial growth factor (VEGF) signaling, essential for proliferation of vascular smooth muscle cells and endothelial cells, respectively[15, 16]. Rhee and colleagues have also shown that another cytosolic 2-Prx isoform, PrxI, regulates the Src kinase and mitosis in the caveolae membrane microdomain and nuclear centrosome, respectively[17, 18]. Hence, the Prx enzymes seem to have the multifaceted roles in cellular ROS detoxification and signal transduction[19]. In this study, we address the pro-tumorigenic function of PrxII in the intestine by investigating a spontaneous intestinal mouse model and human CRC cells. We found that PrxII absence reduced oncogenic β-catenin in the adenomatous polyps as well as the CRC cell with APC mutations. This unexpected result is due to the Axin1-dependent β-catenin degradation enhanced by a $H_2O_2$-dependent inactivation of TNKS1 PARP activity in the absence of PrxII. We further demonstrate a novel redox mechanism by which a zinc-binding motif essential for the PARP activity of TNKS is vulnerable to oxidation and requires the PrxII-dependent antioxidant shielding effect. Finally, the tumor xenograft experiments imply that PrxII inhibitor can be a new therapeutic weapon for combating with CRC.

## Results

**PrxII is essential for APC-mutation-driven intestinal tumorigenesis in vivo.** Although 2-Cys Prxs are ubiquitously expressed

in most tissues, including intestines[20], we found that, by examining the expression pattern of Prx isoforms in the Human Proteome Atlas, PrxII is the most abundant isoform in CRC tissues[21]. In order to examine the CRC-specific function of PrxII in vivo, we generated double-mutant mice by mating $PrxI^{+/-}$ and $PrxII^{+/-}$ mice with $APC^{Min/+}$ mice, which develop multiple intestinal neoplasia (Min) by APC truncation mutation (Supplementary Fig. 1a–c). Although the APC mutation is heterozygous, the intestinal adenomatous polyposis is known to be induced by loss of the residual APC wild-type (WT) copy and thus the resulting adenomatous polyps contain a truncated APC protein similar to those in human colorectal tumors[22]. The small intestines and colons were excised from 12-week-old mice, and intestinal polyps were counted using a stereoscopic microscope (Fig. 1a). The mean number of visible polyps (>0.3 mm in diameter) in the small intestines and colons of $APC^{Min/+};PrxII^{-/-}$ mice was reduced by ~50% compared to those in $APC^{Min/+};PrxII^{+/+}$ and $APC^{Min/+};PrxII^{+/-}$ littermates (Fig. 1b). Histological reviews of small and large intestines revealed that PrxII deletion did not alter the villus structure but decreased the frequency and size of the adenomatous polyps (Fig. 1c). Consequently, $APC^{Min/+};PrxII^{-/-}$ mice (mean survival=241 days) survived much longer than their $APC^{Min/+};PrxII^{+/+}$ (mean survival=146 days) and $APC^{Min/+};PrxII^{+/-}$ (mean survival=152 days) littermates (Fig. 1d). By contrast, the mean number of intestinal polyps in $APC^{Min/+};PrxI^{-/-}$ mice was the same as those in $APC^{Min/+};PrxI^{+/+}$ and $APC^{Min/+};PrxI^{+/-}$ littermates (Supplementary Fig. 1d and e). These data demonstrated that PrxII, not PrxI, promotes intestinal tumorigenesis induced by APC mutation in vivo. We then compared the levels of β-catenin and its target gene expression between polyps from $APC^{Min/+};PrxII^{+/+}$ and $APC^{Min/+};PrxII^{-/-}$ mice. Immunoblot analyses showed that the levels of β-catenin and its transcriptional targets, c-Myc and Cyclin D1, were markedly reduced in polyps from $APC^{Min/+};PrxII^{-/-}$ mice compared to those in polyps from $APC^{Min/+};PrxII^{+/+}$ mice (Fig. 1e). Unexpectedly, the level of Axin1, a key scaffold protein in β-catenin destruction complex, was inversely increased in polyps from $APC^{Min/+};PrxII^{-/-}$ mice. Since the mRNA levels of β-catenin and Axin1 were unchanged between polyps from $APC^{Min/+};PrxII^{+/+}$ and $APC^{Min/+};PrxII^{-/-}$ mice (Supplementary Fig. 1f), our data suggest that PrxII regulates Axin1 and β-catenin at protein level in vivo. Given that the β-catenin target genes are involved in proliferation and survival of CRC cells[23], we counted the proliferating and dead cells in the polyps. Polyps from $APC^{Min/+};PrxII^{+/+}$ and $APC^{Min/+};PrxII^{-/-}$ mice contained similar proportions of proliferating cells, as assessed by the Ki-67 expression and BrdU incorporation (Supplementary Fig. 2a and b). In addition, the BrdU incorporation assays also showed that PrxII deletion had absolutely no effect on proliferation and migration of intestinal epithelial cells in crypts (Supplementary Fig. 2c). On the contrary, the number of dead cells measured by TUNEL staining was markedly higher in polyps from $APC^{Min/+};PrxII^{-/-}$ mice than in those from $APC^{Min/+};PrxII^{+/+}$ littermates (Fig. 1f). Thus, the in vivo data reveal that PrxII promotes the survival of tumor cells in the intestinal adenomatous polyposis driven by APC mutation.

**PrxII promotes the tumorigenic activity of APC-mutant CRC cells via Axin-β-catenin axis.** We then analyzed the mechanisms of regulation of β-catenin levels by PrxII in a panel of human CRC cell lines highly expressing PrxII. Specific knockdown of PrxII expression by different small-interfering RNAs (siRNAs) markedly reduced the levels of endogenous β-catenin in SW480 and HT29 cells, both expressing truncated mutant forms of APC protein (Fig. 2a). However, PrxI depletion did not alter β-catenin

level in these CRC cells. More importantly, the decline in the β-catenin level was proportional to the extent of PrxII depletion (Fig. 2b), suggesting that stringent PrxII knockdown is crucial for reducing β-catenin level. PrxII depletion also reduced the levels of

total and, more importantly, active (unphosphorylated form) β-catenin in other APC-mutant CRC cells, SW620, DLD-1, and CoLo205 (Fig. 2c). To exclude off-target effects of PrxII siRNAs, PrxII expression was rescued by transfection of HT29 cells with

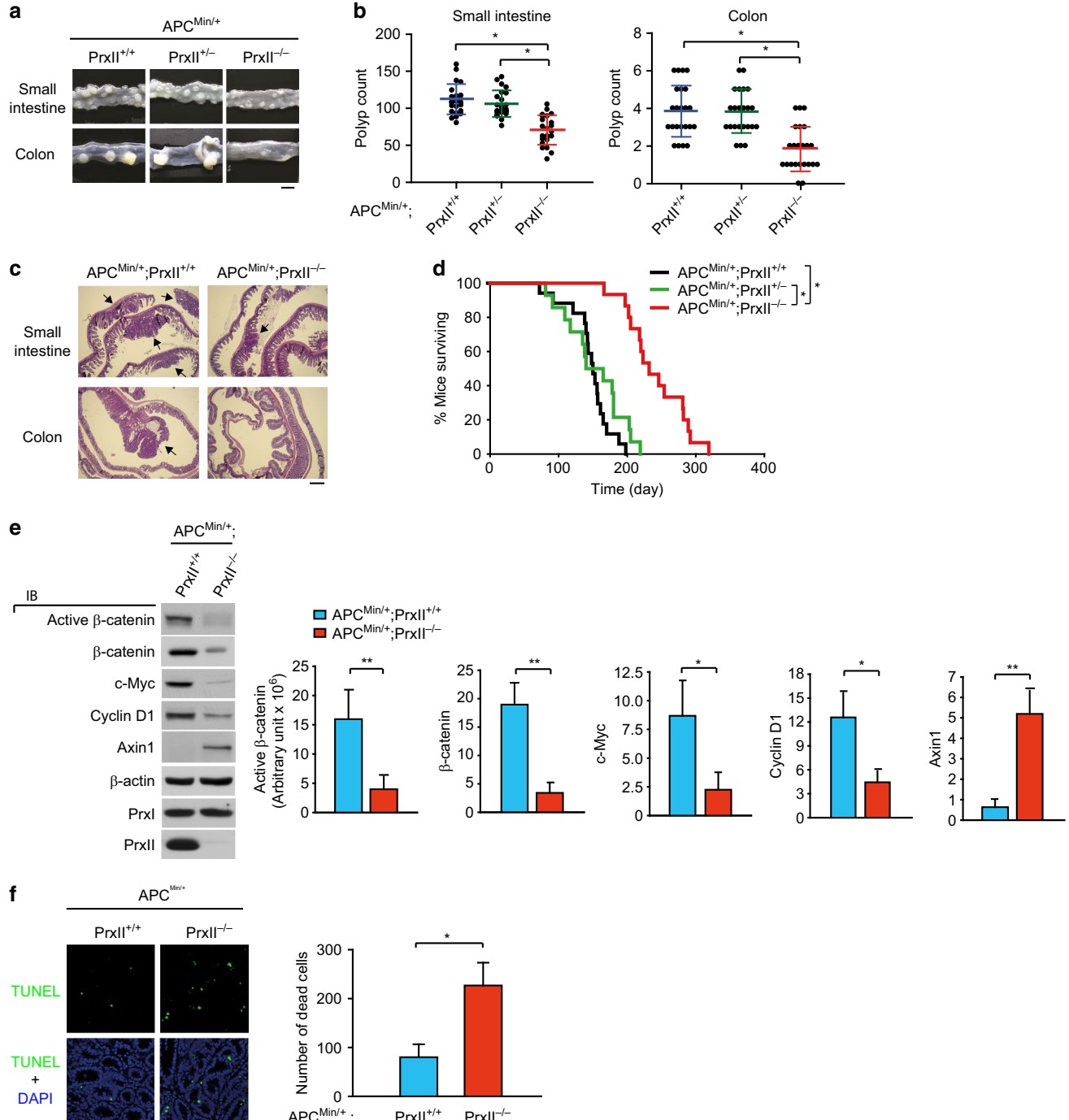

**Fig. 1** Loss of PrxII suppresses intestinal tumorigenesis and increases survival in APC-mutant mice. **a** Representative photomicrographs of longitudinally opened intestinal tissues from $APC^{Min/+};PrxII^{+/+}$, $APC^{Min/+};PrxII^{+/-}$, and $APC^{Min/+};PrxII^{-/-}$ mice. Scale bar, 5 mm. **b** Macroscopic polyps (>3 mm in diameter) were counted in whole-mount intestinal tissues from $APC^{Min/+};PrxII^{+/+}$, $APC^{Min/+};PrxII^{+/-}$, and $APC^{Min/+};PrxII^{-/-}$ mice. Dot plots represent the average polyp numbers ± s.d. ($n = 23$ mice per group, $*P < 0.001$ with repeated-measures ANOVA). **c** Cross sections of frozen small intestines and colons from $APC^{Min/+};PrxII^{+/+}$ and $APC^{Min/+};PrxII^{-/-}$ mice were stained by hematoxylin and eosin. Arrows indicate the polyps. Scale bar, 1 mm. **d** Kaplan–Meier plot showing the survival of $APC^{Min/+};PrxII^{+/+}$ ($n = 17$), $APC^{Min/+};PrxII^{+/-}$ ($n = 14$), and $APC^{Min/+};PrxII^{-/-}$ ($n = 15$) mice ($*P < 0.001$ with Log rank test). **e** Intestinal polyps from $APC^{Min/+};PrxII^{+/+}$ and $APC^{Min/+};PrxII^{-/-}$ mice were homogenized and immunoblotted (IB) for indicated proteins. Data in the graph are means ± s.d. of the intensities of immunoreactive bands after being normalized with control β-actin level ($n = 3$, $*P < 0.05$, $**P < 0.01$). **f** Cell death in the intestinal polyps from $APC^{Min/+};PrxII^{+/+}$ and $APC^{Min/+};PrxII^{-/-}$ mice was measured by TUNEL staining. Nuclei were stained with DAPI. Data in the graph are means ± s.d. of the number of TUNEL-positive cells per mm$^2$ ($n = 9$ mice per group, $*P < 0.01$). Scale bar, 50 μm

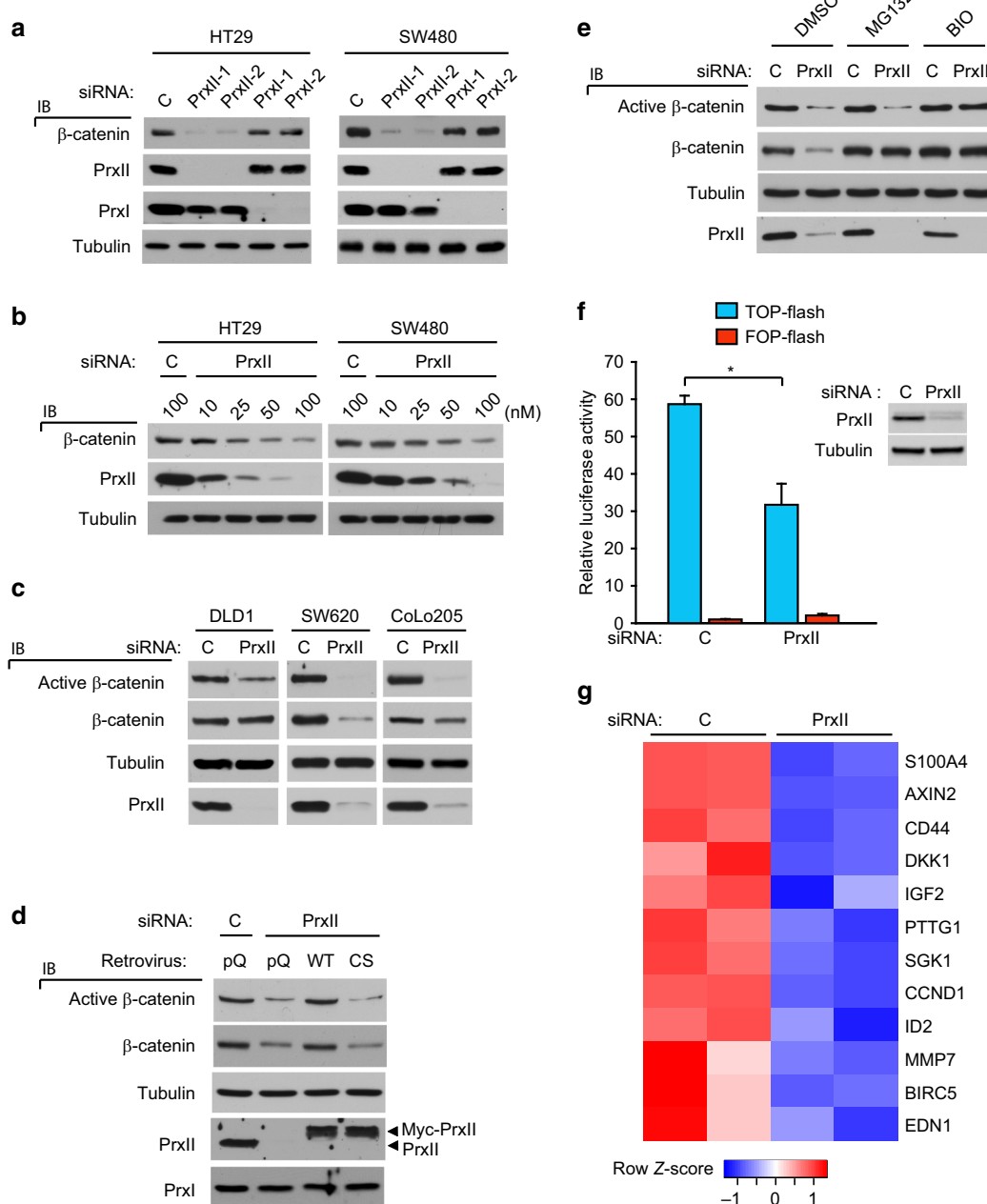

**Fig. 2** PrxII is required for stabilization of oncogenic β-catenins in APC-mutant CRC cells. **a** and **b** HT29 and SW480 cells were transfected with a set of specific siRNAs against PrxI and PrxII **a** or different concentrations of PrxII-1 siRNA **b** for 48 h and immunoblotted (IB) for total β-catenin. **c** DLD1, SW620, and CoLo205 CRC cells were transfected with control or PrxII-1 siRNA and immunoblotted for indicated proteins. Active β-catenin was immunoblotted with antibody specifically recognizing β-catenin unphosphorylated on S37/T41. **d** HT29 cells were transfected with control or PrxII-1 siRNA for 24 h and then infected with retroviruses expressing siRNA-resistant PrxII WT and C51/172S mutant (CS) for additional 24 h. Empty retrovirus (pQ) was used for control. The exogenously expressed PrxII is tagged at C-terminus with a Myc epitope. **e** Control and PrxII-depleted SW480 cells were treated with proteasome and GSK3β inhibitors (25 μM MG132 and 2.5 μM BIO) for 1 h. Control vehicle was dimethyl sulfoxide (DMSO). **f** Control and PrxII-depleted SW480 cells were transfected with FOPflash or TOPflash luciferase reporter plasmid for measuring β-catenin/TCF transcription activity. Luciferase activity was measured in triplicate by dual-luciferase assay. Data in the graph are means ± s.d. of relative luciferase activities after normalization of transfection efficiency ($n = 3$, *$P < 0.002$). **g** Heat map depicting the significant downregulation of β-catenin/TCF-dependent gene expression by PrxII depletion in HT29 cells. Duplicate experiments are shown in parallel. A representative immunoblot (IB) from three independent experiments is shown. Firefly luciferase-specific siRNA was used as control (C)

siRNA-resistant forms of PrxII. The PrxII WT perfectly restored levels of both total and active β-catenins to those comparable with control siRNA-transfected cells, indicating the specific role of PrxII in regulating β-catenin level (Fig. 2d and Supplementary Fig. 3a). In contrast, the expression of peroxidase-inactive mutants (C172S and C51/172 S) of PrxII[24] did not restore β-

catenin levels, indicating that the peroxidase activity of PrxII is necessary for maintaining active β-catenin levels in CRC cells. Indeed, PrxII depletion increased the intracellular H$_2$O$_2$ level in HT29 and SW480 cells (Supplementary Fig. 3b).

Since we observed that PrxII depletion did not change the β-catenin mRNA level, we considered the degradation of the

β-catenin proteins by the canonical destruction complex carrying sequential phosphorylation of β-catenin by casein kinase-1 and glycogen synthase kinase (GSK)-3β[25]. The phosphorylated β-catenins are ubiquitinated by β-TrCP ubiquitin E3 ligase and subsequently degraded by proteasome. Thus, we examined whether canonical β-catenin destruction complex is involved in the PrxII-dependent regulation of β-catenin level. In HT29 and SW480 CRC cells, proteasome and GSK3β inhibitions blocked the reduction of β-catenin level induced by PrxII depletion. Notably, GSK3β inhibitors increased the active β-catenin level, suggesting that constitutively active GSK3β was involved (Fig. 2e and Supplementary Fig. 4a–c). However, we excluded the possibility that PrxII depletion might stimulate GSK3β activation by examining the level of tyrosine phosphorylation of GSK3β, which is an indicative of the kinase activation[26] (Supplementary Fig. 4d). In addition, we observed that PrxII depletion had no effect on Wnt-induced β-catenin stabilization (Supplementary Fig. 4e), confirming that PrxII selectively participates in deregulated β-catenin signaling. As the consequence of the reduced β-catenin level, PrxII depletion elicited a marked decrease in TCF-dependent reporter expression (Fig. 2f). We then examined the β-catenin/TCF-dependent transcription in HT29 cells by mRNA sequencing (Supplementary Fig. 5a and b). In particular, PrxII depletion downregulated 12 genes (FDR<0.05) among the β-catenin target genes being expressed in CRC cells[27], where major β-catenin target genes, such as CCND1, AXIN2, and BIRC5, were included (Fig. 2g). It is noteworthy that other metastasis- and cell cycle-promoting genes, such as S100A4, MMP7, ID2, and PTTG1, were also downregulated by PrxII depletion. In addition to HT29 cells, the mRNA sequencing also revealed that PrxII depletion downregulated thirteen β-catenin target genes (FDR<0.05) in SW480 cells, among which four genes (CCND1, S100A4, ID2, and EDN1) were overlapped (Supplementary Fig. 5c). It is, however, noted that downregulation of several other genes in SW480 cell might be mediated by a distinct β-catenin-associated transcription complex[28] or a secondary effect of β-catenin reduction. Therefore, the data demonstrate that PrxII absence accelerates the degradation of transcriptionally active β-catenins via the canonical destruction complex in CRC cells.

Subsequently, we questioned how PrxII depletion induces the Wnt-independent activation of β-catenin destruction complex in CRC cells. When APC is mutated, another scaffold protein Axin1 becomes important for the β-catenin destruction[29, 30]. Thus, its over-expression in APC-mutant CRC cells sufficiently induces β-catenin degradation[31]. In fact, we previously observed the increased level of Axin1 in the intestinal polyps from PrxII knockout mice. We, therefore, examined the level of Axin1 and Axin1-associated destruction complex in CRC cells. An immunoblot analysis showed that PrxII depletion increased the levels of endogenous Axin1 proteins in both HT29 and SW480 CRC cells, which was inversely correlated with the active β-catenin levels (Fig. 3a). A subsequent co-immunoprecipitation (IP) experiment showed that PrxII depletion significantly augmented the abundance of the Axin1-associated destruction complexes (Fig. 3b). When GSK3β inhibitor BIO was treated, the phospho-β-catenins disappeared from the complexes. Moreover, knockdown of Axin1/2 restored active β-catenin levels in PrxII-depleted cells to a level in control cells (Fig. 3c). These results indicate that Axin orchestrated β-catenin degradation by forming functional destruction complexes in the PrxII-depleted CRC cells. Since Axin is degraded by the poly(ADP-ribose) polymerization (PARsylation) and subsequent ubiquitination[32, 33], we analyzed the status of Axin1 in HT29 and SW480 cells. Indeed, treatment of the proteasome inhibitor MG132 induced the accumulation of PARsylated and ubiquitinated Axin1 proteins in control cells,

indicating that Axin1 is constantly degraded. By contrast, PrxII depletion blocked PARsylation/ubiquitination of Axin1 without affecting total ubiquitination (Fig. 3d). Then, we evaluated the biological significance of PrxII-dependent regulation of Axin1/β-catenin pathway in CRC cells by examining the colony-forming ability. The in vitro colony formation assay showed that PrxII depletion sufficiently inhibited the colony formation of HT29 and SW480 cells, not RKO cells expressing APC WT (Fig. 3e). More strikingly, the expression of constitutively active β-catenin S37A mutant almost completely rescued the colony-forming ability of APC-mutant CRC cells that had been impaired by PrxII depletion (Fig. 3f). Taken together, the results indicate that PrxII absence can sufficiently reverse the oncogenic phenotype of APC mutation by inducing Axin1-directed β-catenin degradation.

**PrxII protects TNKS from $H_2O_2$-dependent inactivation in APC-mutant CRC cells.** Since TNKS is a sole PARsylating enzyme for Axin proteins[4], we examined whether PrxII is essential for the TNKS activity by performing the in vitro PARP assay. Surprisingly, PrxII depletion severely impaired TNKS activity in the APC-mutant HT29 and SW480 cells but not in the APC-competent RKO cells (Fig. 4a). By contrast, $H_2O_2$ treatment inhibited the TNKS activity in both SW480 and RKO cells (Fig. 4b), which indicates that regardless of APC mutations, exogenous $H_2O_2$ directly inactivated TNKS activity. As TNKS is known to be auto-PARsylated and degraded[34], we examined the levels of TNKS along with its substrate proteins in a panel of CRC cells. Indeed, the PrxII depletion increased the levels of TNKS and Axin1 proteins in all APC-mutant CRC cells we tested, but not in APC-competent cells including RKO and Colo741 cells (Fig. 4c). More importantly, PrxII depletion did not affect the level of telomeric repeat-binding factor 1 (TRF1) protein, another TNKS substrate essential for telomerase regulation in the nucleus. By contrast, direct inhibition of TNKS using a specific inhibitor XAV939[32] increased the levels of TNKS and both substrates, Axin1 and TRF1, in all CRC cells, and it consequently inhibited proliferation of both HT29 and SW480 cells (Supplementary Fig. 6). To ascertain a direct correlation between APC and PrxII function in CRC cells, we performed APC knockdown in RKO cells. APC knockdown certainly induced a marked increase in levels of cellular $H_2O_2$ and β-catenins (Fig. 4d and e) and consequently accelerated the $H_2O_2$-dependent inactivation of TNKS1 (Fig. 4f). Moreover, combination of APC and PrxII knockdown increased the levels of TNKS and Axin1 proteins accompanied by a decrease in β-catenin levels. Given that PrxII is a cytosolic peroxidase, the results suggest that PrxII might selectively protect TNKS in the cytosol from oxidative stress induced by APC mutation or loss.

As a molecular mechanism underlying $H_2O_2$-mediated inactivation of the PARP activity of TNKS, we anticipated the presence of oxidation-sensitive Cys residues within the PARP catalytic domain in TNKS. To search for it, we aligned the peptide sequences of various PARP domains and found that five Cys residues, including three in a zinc-binding motif[35], are uniquely present on TNKS isoforms among PARP family members. As a result, we mutated each Cys residue to Ser in TNKS1 and examined its PARP activity. Unexpectedly, mutations of the three zinc-coordinating Cys residues (C1234, C1242, and C1245), but not the other two Cys residues, resulted in a complete loss of PARP activity (Fig. 4g). To test whether zinc-binding motif was labile under oxidative conditions, we prepared a recombinant TNKS1 PARP domain (amino acids 1023–1327). The TNKS1 PARP domain showed intact PARsylating activity, which was completely inactivated by incubation with $H_2O_2$ (Fig. 4h). We then postulated that a zinc ion might be popped out upon

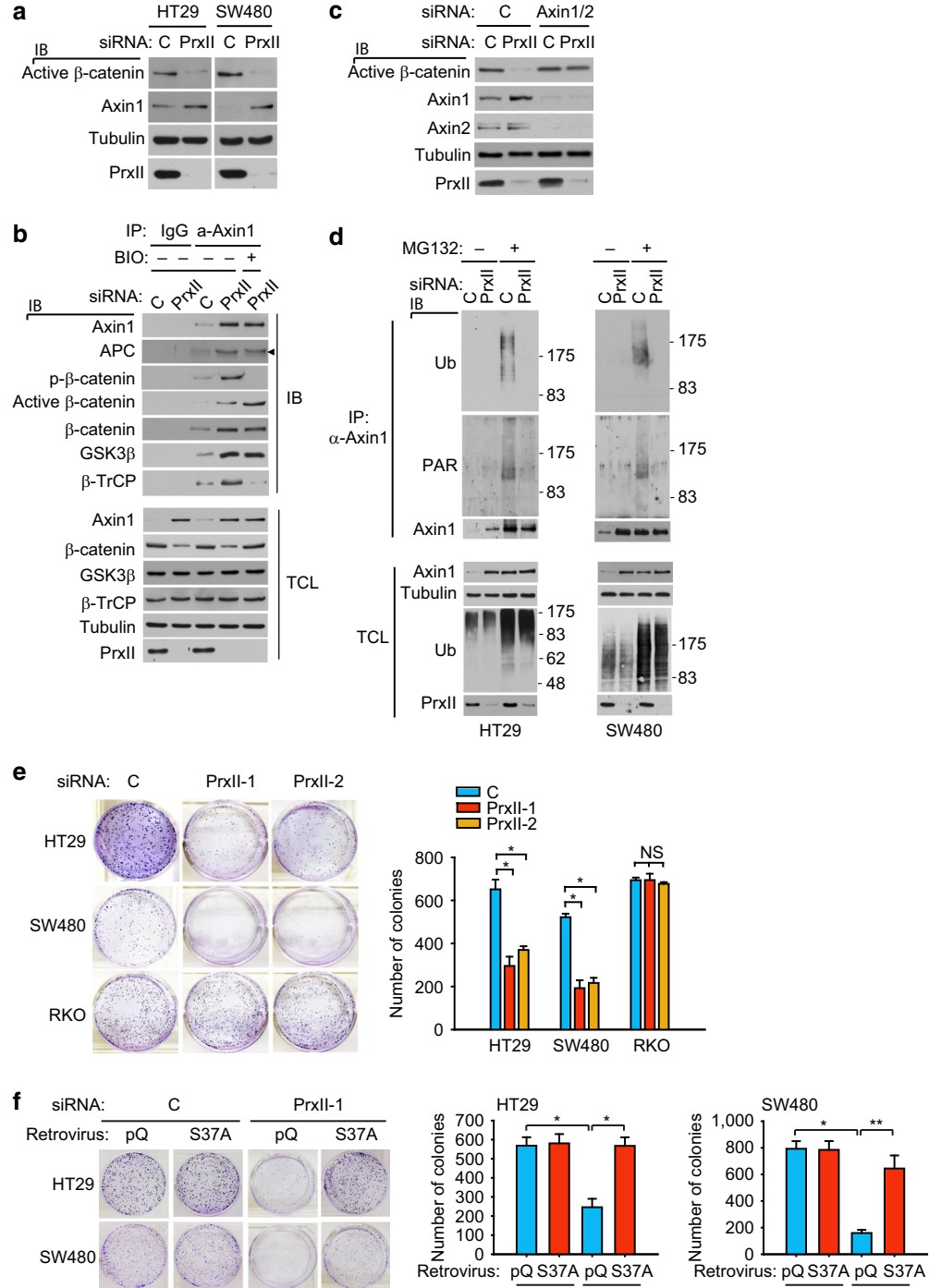

Fig. 3 PrxII promotes poly(ADP) ribosylation-dependent Axin1 degradation and tumorigenic activity in APC-mutant CRC cells. **a** HT29 and SW480 cells were transfected with control or PrxII-1 siRNA and immunoblotted (IB) for active β-catenin and Axin1. **b** SW480 cells were transfected with control or PrxII-1 siRNA and lysed for immunoprecipitation (IP) with control IgG or anti-Axin1 antibody. BIO was treated to SW480 cells for 1 h before lysis. The IP complexes were immunoblotted for components of β-catenin destruction complexes. Arrowhead indicates a truncated APC protein. **c** SW480 cells were co-transfected with indicated siRNAs including PrxII-1 and Axin1/2-specific siRNAs and immunoblotted for active β-catenin. **d** HT29 and SW480 cells were transfected with control or PrxII-1 siRNA and lysed for IP with anti-Axin1 antibody. MG132 was treated to SW480 cells for 1 h before lysis in order to accumulate ubiquitinylated proteins. The IP complexes were immunoblotted for PARsylation (PAR) and ubiquitination (Ub). The molecular weight markers are labeled in kilodaltons. **e** RKO, HT29, and SW480 cells were transfected with each of two PrxII siRNAs and then cultured for 10 days. The colonies were stained with crystal violet and counted. Data in the graph are the average number of colonies ± s.d. ($n = 3$, *$P < 0.001$). NS, statistically not significant. **f** HT29 and SW480 cells expressing β-catenin S37A mutant were transfected with PrxII-1 siRNA and then cultured for 10 days. The colonies were stained with crystal violet and counted. Data in the graph are the average number of colonies ± s.d. ($n = 3$, *$P < 0.001$, **$P < 0.005$). All experiments were independently repeated three times. Total cell lysates (TCL) were immunoblotted as control for the amount of indicated proteins in the lysate. Firefly luciferase-specific siRNA was used as control (C)

cysteine oxidation, as previously shown in the examples of zinc-binding proteins[36, 37]. The released free zinc ions can be measured spectrophotometrically using a 4-(2-pyridylazo) resorcinol[38]. Using this method, we demonstrated that $H_2O_2$ treatment induced almost complete release of zinc ions, such that more than 90% of TNKS1 PARP proteins had ultimately lost zinc ions (Fig. 4i). The data indicate that the zinc-binding motif in TNKS is catalytically essential for PARP activity and that $H_2O_2$-induced oxidation of Cys residues induces the liberation of zinc ions from the PARP domains. Together, these results constitute strong evidence for the redox regulation of TNKS activity by PrxII-regulated $H_2O_2$.

**PrxII is a novel TNKS-interacting protein in the APC-mutant CRC cells.** To address the key question of how PrxII specifically protects TNKS from the $H_2O_2$-mediated inactivation, we investigated the protein–protein interaction between PrxII and TNKS. Co-IP experiments showed that endogenous TNKS interacted with PrxII only in APC-mutant HT29 and SW480 cells, not in APC-competent RKO cells (Fig. 5a). By contrast, TNKS did not interact with PrxI in either cell type, which confirms the specific role of PrxII in the TNKS/Axin1/β-catenin pathway. To characterize the direct interaction between TNKS1 and PrxII, we ectopically over-expressed two proteins in human embryonic kidney cells HEK293 as a non-CRC cell. As a result, co-IP

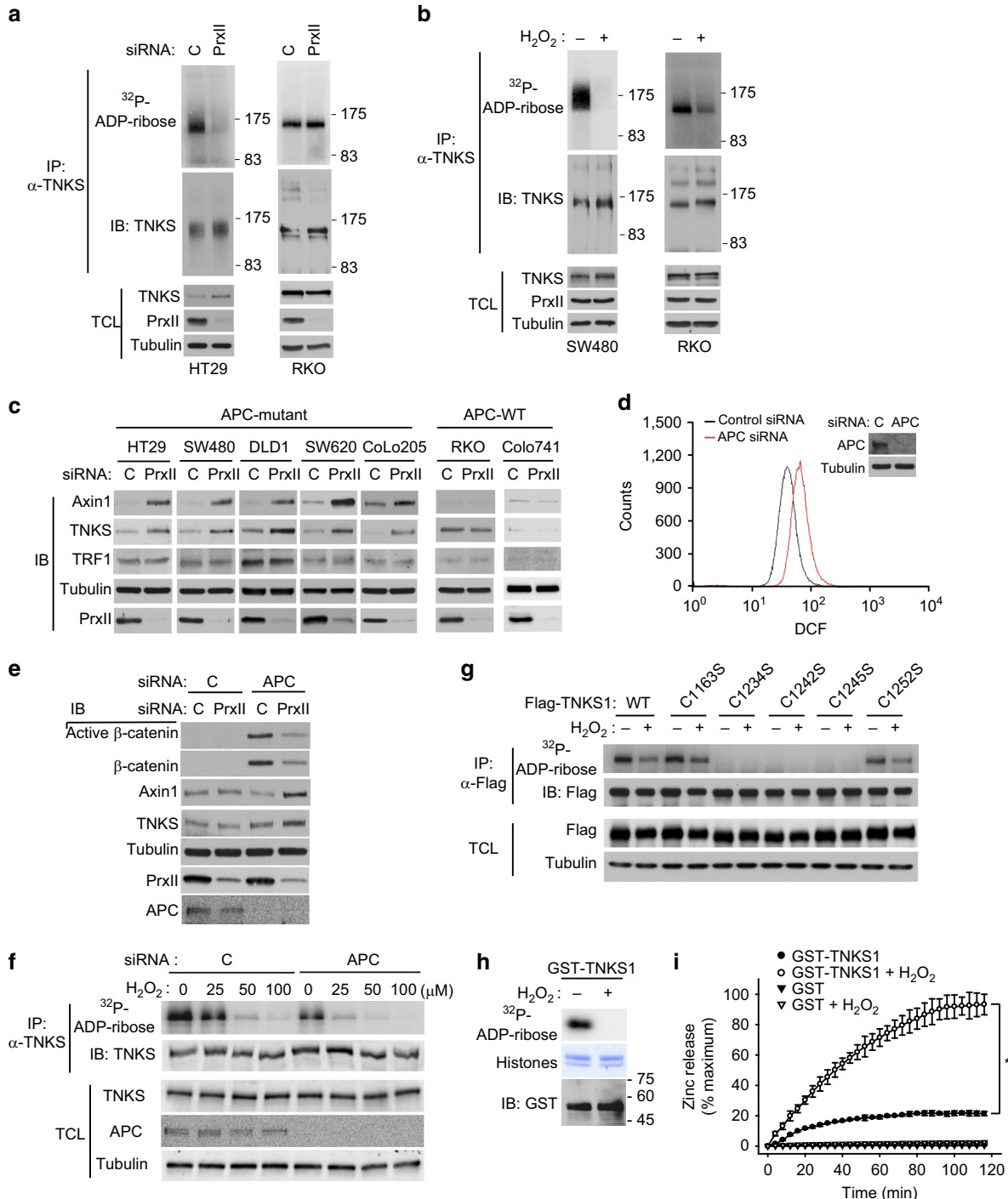

experiments indicated that TNKS and PrxII interacted directly (Fig. 5b). More definitely, an in situ proximity ligation assay (PLA) visualized that direct interaction between TNKS and PrxII occurred in the cytoplasm of HT29 and SW480 cells, not RKO cells (Fig. 5c and Supplementary Fig. 7). Red fluorescence signals representing interaction of two proteins almost completely disappeared by PrxII depletion. Thus, the data together indicate that redox shielding TNKS by PrxII is highly specific by protein–protein interaction and is dependent on *APC* mutation. Subsequently, we analyzed the molecular interaction map by performing a series of TNKS and PrxII mutagenesis. When truncated TNKS mutants were co-expressed with PrxII, the IP experiments showed that PrxII interacted with the ankyrin repeat cluster (ARC) 4/5 domains of TNKS (Fig. 5d). It is notable that PrxII binding is not overlapped with Axin1, which binds to ARC2/3 domains[39]. On the other hand, the TNKS ARC domains recognize the consensus sequence RXXPXG in client proteins, where the Gly residue at position 6 is crucial for direct binding[40]. We thus searched for similar hexapeptide sequences on PrxII and introduced Gly-to-Val mutations to the three potential consensus regions. Co-IP experiments showed that only G116V mutation among the three mutated sites completely abolished binding of PrxII to TNKS1 (Fig. 5e). Although the PrxII WT and G116V mutant exhibited the same level of peroxidase activity (Supplementary Fig. 8), the PrxII G116V mutant did not prevent the inhibition of TNKS activity by $H_2O_2$, whereas PrxII WT did (Fig. 5f). In addition, the Prx-$SO_{2/3}$ blot indicated that exogenous PrxII with a C-terminal Myc tag (PrxII-Myc) was resistant to hyperoxidation, while endogenous 2-Cys Prxs were fully hyper-oxidized, upon $H_2O_2$ treatment (lanes 2 vs. 3 in Fig. 5f). Indeed, the in vitro Prx activity assay using recombinant enzymes demonstrated that, unlike intact PrxII enzyme, the PrxII-Myc enzyme exhibited a strong peroxidase activity without any indication of hyperoxidation (Supplementary Fig. 8c). Since previous studies have shown that the C-terminal modification on PrxII protein confers resistance to hyperoxidation[41–43], we speculate that addition of Myc tag to C-terminus may cause a similar structural change on PrxII, which then converts to hyperoxidation-resistant form. We also performed the colony formation assay to determine the biological significance of PrxII–TNKS interaction. The colony formation of APC-mutant SW480 cells inhibited by PrxII depletion was completely rescued by the ectopic expression of PrxII WT but not G116V mutant (Fig. 5g). Collectively, these results indicate that bound PrxII may prevent the oxidative inactivation of tankyrase by eliminating vicinal $H_2O_2$, which is crucial for the propagation of APC-mutant CRC cells.

**Chemical inhibition of PrxII kills the APC-mutant CRC cells in vitro and in vivo.** The gene expression analysis in The Cancer Genome Atlas database[1] confirmed that *PrxII* expression was significantly higher in tumor specimens from patients with colon adenocarcinoma than in normal colon tissues, whereas the expression of the closest isoform *PrxI* showed no such difference (Fig. 6a). The elevated *PrxII* expression was observed at all tumor stages (Supplementary Fig. 9a). Consistently, our immunohistochemistry using CRC tissue array also showed that PrxII level was about two-fold higher in the CRC tissues compared to the normal colon tissues (Fig. 6b). These results suggests that specific *PrxII* induction may be a prerequisite for the CRC expansion. Based on the expression analyses, we performed a proof-of-concept experiment using a previously identified chemical inhibitor to assess the therapeutic potential of PrxII inhibition relevant to human CRC. To our knowledge, there have been no isoform-specific chemical compounds that inhibit the peroxidase activity of human PrxII. We, therefore, decided to test a cell-permeable compound called conoidin A that was shown to covalently bind to a parasite PrxII[44]. By performing the in vitro Prx assay, we found that conoidin A inhibited human PrxII activity by about 85% ($214.9 \pm 24.3$ nmol per min for control vehicle vs. $38.8 \pm 17.1$ nmol per min for conoidin A treatment) and also inhibited the PrxI activity by about 50% (Fig. 6c). It is, however, interesting that the apparent initial rate of PrxI activity was unaffected by conoidin A ($263.7 \pm 45.4$ nmol per min for control vehicle vs. $241.7 \pm 21$ nmol per min for conoidin A), which is explained by the previous indication that conoidin A promotes the hyperoxidation of PrxI[45]. Then, colony formation assay showed that conoidin A treatment sufficiently inhibited the proliferation of HT29 and SW480 cells, not RKO cells, which implicates a therapeutic potential of PrxII inhibition to selectively target human APC-mutant CRC cells (Fig. 6d). When conoidin A was injected intraperitoneally to mice bearing HT29-derived tumor xenografts, conoidin A treatment significantly retarded the tumor growth compared to the control treatment (Fig. 6e and f); the result supports that PrxII inhibition can be a novel targeted therapy for human CRC.

## Discussion

There are only few targeted therapies, such as inhibition of VEGF and epidermal growth factor receptor, available for the advanced CRC patients[46, 47]. Although TNKS has emerged as a new option for targeted CRC therapy, there is concern that inhibition of TNKS may cause pleiotropic effects owing to its broad substrates[4]. Moreover, the intrinsic mechanisms regulating TNKS

**Fig. 4** PrxII is required for PARP activity of TNKS that is sensitive to oxidative inactivation by $H_2O_2$. **a** HT29 and RKO cells were transfected with control or PrxII-1 siRNA and lysed for immunoprecipitation (IP) with anti-TNKS antibody. The immunoprecipitated TNKS was subjected to in vitro PARP assay as described in the Methods. Total cell lysate (TCL) was immunoblotted for checking equal use of proteins. The molecular weight markers are labeled in kilodaltons. **b** SW480 and RKO cells were treated with or without $H_2O_2$ (final conc. 100 μM) for 30 min. The immunoprecipitated TNKS was subjected to *in vitro* PARP assay. The molecular weight markers are labeled in kilodaltons. **c** CRC cells were transfected with control or PrxII-1 siRNA and then immunoblotted (IB) for indicated proteins. **d** Intracellular $H_2O_2$ level was measured in RKO cells that had been transfected with control or APC siRNA. The cells were incubated with 5,6-chloromethyl-2′,7′-dichlorodihydrofluorescein diacetate at 37 °C for 1 h and then the dichlorofluorescein (DCF) fluorescence was analyzed with a flow cytometry. A representative cell sorting datum of three experiments is shown. **e** RKO cells were transfected with either APC siRNA alone or APC plus PrxII-1 siRNAs and then immunoblotted for indicated proteins. **f** RKO cells were transfected with control or APC siRNA and then treated with increasing concentrations of $H_2O_2$ for 10 min. The immunoprecipitated TNKS was subjected to in vitro PARP assay. **g** HEK293 cells were transfected with plasmid vectors encoding TNKS1 WT and CS (Cys-to-Ser substitution) mutants and treated with or without 100 μM $H_2O_2$ for 10 min. The expressed Flag-TNKS1 enzymes were immunoprecipitated with anti-M2 (Flag) antibody and subjected to in vitro PARP assay. **h** Recombinant TNKS1-PARP proteins (aa 1023–1327) in *E. coli* extract were pooled down with glutathione-Sepharose 4B beads, incubated for 10 min in the presence or absence of 100 μM $H_2O_2$, and then subjected to in vitro PARP assay. The histone mixture (Coomassie blue staining) was used as substrate. **i** Purified recombinant TNKS1-PARP proteins were incubated with 500 μM $H_2O_2$ for zinc releasing assay as described in the Methods. Data in the graph are the average percent increases ± s.d. of free zinc ions vs. maximum zinc content in the purified GST-TNKS1-PARP protein used for the reaction ($n = 3$, *$P < 0.001$ with repeated measures ANOVA). Immunoblots (IB) or $^{32}$P-autoradiographs shown are a representative of three independent experiments. Total cell lysates (TCL) were immunoblotted as control for the amount of indicated proteins. Firefly luciferase-specific siRNA was used as control (C)

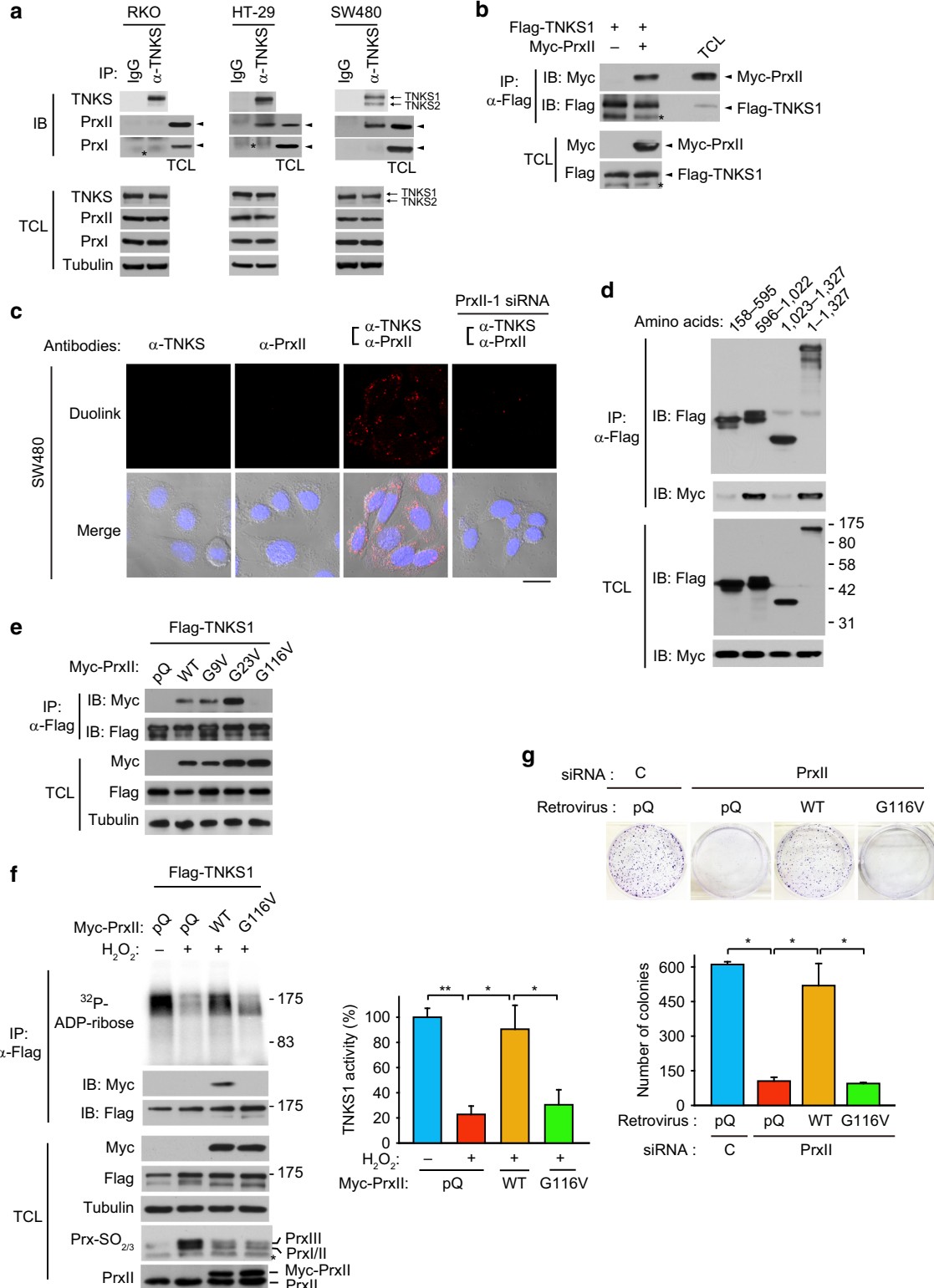

activity in intestinal tumorigenesis are largely unknown. Our study demonstrates for the first time that a zinc-binding motif in TNKS is essential for PARP activity and is protected from oxidative inactivation by the redox enzyme PrxII in CRC cells with endogenous $H_2O_2$ level heightened by *APC* mutation (Fig. 7). In particular, our data emphasize that the PrxII-dependent regulation of TNKS activity is highly specific and unique in several aspects: (1) PrxII is the first redox enzyme

directly bound to TNKS via ARC4/5 domains; (2) PrxII selectively targets TNKS in cytosol; and (3) function of PrxII in CRC is specific to deregulated β-catenin pathway (e.g., *APC* mutation).

Prx family consists of six isozymes divided by 2-Cys and 1-Cys subfamilies. The 2-Cys subfamily enzymes are thioredoxin-dependent peroxidases broadly conserved from bacteria to human[42, 48]. Among the 2-Cys Prxs, cytosolic PrxI and PrxII isozymes are overexpressed in various cancer types and known to

play important regulatory roles in the membrane receptor-mediated signal transduction[10]. They are thought to be redundant because the subcellular distribution and physical properties, such as catalytic efficiency and protein structure, are almost identical. However, the differential phenotypes of knockout mice are notable: PrxI deficiency results in the malignant tumors in aged mice, whereas PrxII deficient mice live normally albeit a defect in red blood cells[49, 50]. In this study, we compared the in vivo functions of PrxI and PrxII in the mouse model of *APC* mutation-induced intestinal tumorigenesis and demonstrated the specific pro-tumorigenic role of PrxII only. We identified the TNKS-ARC binding region (R[110]LSEDYG[116]) on PrxII. However, PrxI has the alternative residues within the corresponding consensus sequence (T[111]IAQDYG[117]). Since additional co-IP experiment using PrxI-T111R mutant showed that a single substitution of Thr to Arg at position 1 was insufficient to induce the interaction with TNKS1, we anticipate that the configuration of entire ARC-binding region is perhaps involved in the interaction with TNKS. It has been shown that PrxI is preferential for molecular chaperone rather than peroxidase upon oxidative stress due to the presence of additional Cys83 residue, which is not present in PrxII[51]. Thus, our protein–protein interaction data provide another molecular evidence that minor variations in amino acid sequence may cause differential cellular functions among Prx isoforms.

In general, somatic mutations on oncogenes and tumor suppressors cause intrinsic oxidative stress in cancer cells by amplifying the ROS production. For example, activating mutations of oncogenes, such as *Hras* and *c-Myc*, are known to induce a robust induction of intracellular ROS production[52, 53]. Inactivation of p53 tumor suppressor, which regulates the expression of various antioxidants, is also known to increase the cellular ROS level[54]. On the contrary, there was evidence that expression of activated oncogenes, such as *KRas*[G12D], *B-Raf*[V619E], and *c-Myc*[ERT2], increases the Nrf2 transcription, which then elevates antioxidant systems to reduce cellular ROS level in mouse model and human pancreatic cancer[55]. Such complications may arise from the double-edged function of ROS in cancer cells. A moderate and transient induction of cellular ROS level is undoubtedly required for hyper-proliferation of cancer cells due to a second messenger role in growth factor signaling[56]. On the contrary, excess ROS level is mutagenic and cytotoxic due to the oxidative damages on macromolecules[57]. Likewise, the role of cellular ROS in the intestinal tumorigenesis seems complex. Previously, inactivating *APC* mutation was found to increase cellular superoxide production by Rac1-dependent activation of NADPH oxidase[58]. In that study, the Rac1-dependent superoxide production was required for hyper-proliferation of the transformed intestinal progenitor cells in the crypt. On the contrary, our experiments reveal that $H_2O_2$ level regulated by PrxII is critical for the survival of tumorigenic epithelial cells in mouse intestinal adenomas, as well as in human APC-mutant CRC cells; whereas it is not involved in the proliferation of intestinal progenitor cells in the crypt. Nonetheless, we found that the $H_2O_2$ level was also increased by the APC knockdown in RKO cells. By putting these two complementary studies together, we propose that Rac1-dependent superoxide is required for expansion and transformation of intestinal stem cells in the crypt; whereas PrxII-regulatable $H_2O_2$ intimidates the tumorigenic epithelial cells by targeting TNKS in growing intestinal tumors and is not necessary for classical Wnt/β-catenin signaling.

The most intriguing characteristic of PrxII function in CRC was specificity to APC-mutant context. Our data evidence that the protein–protein interaction between PrxII and TNKS is associated with *APC* mutation. Initially, we anticipated a correlation between PrxII expression and *APC* mutation because PrxII expression is upregulated in CRC. However, the TCGA database analysis revealed that PrxII expression have no significant correlation with *APC* mutations in human CRC (Supplementary Fig. 9b). We also observed that APC truncation mutation and knockdown did not change the PrxII level in mouse intestines and human CRC cells, respectively. Rather, silencing of PrxII expression by promotor hypermethylation have been frequently reported in many other cases of human cancers[59–61]. Thus, we concluded that *APC* mutations are irrelevant to transcriptional regulation of PrxII in CRC. Although the *APC* mutation dependency of PrxII–TNKS interaction was not fully understood, we are currently examining a post-translational modification of PrxII bound to TNKS. By examining the PrxII structure (PDB entry 1QMV)[62], we point out that Gly116 residue locates slightly below the surface in the dimeric structure of PrxII. Since the Gly residue locates far apart from the active-site Cys residues, it was anticipated that the binding to TNKS cannot interfere the peroxidase activity. Indeed, the Gly-to-Val substitution had no effect on PrxII activity. According to the crystal structure of TNKS-Axin1 complex[39], the conserved Gly residue in Axin1 protein lies in the bottom of a narrow groove on the surface of TNKS ARC domain. Thus, we conceive that *APC* mutation may trigger a post-translational modification of PrxII, which in turn causes a conformational change enabling the protrusion of the Gly residue for TNKS interaction. So far, numerous studies including ours have shown the inhibitory regulation of Prx activity by covalent modifications, such as phosphorylation, acetylation, and hyper-oxidation[17, 41, 63]. However, we have failed to see any of the previously known modification on PrxII among CRC cell lines,

**Fig. 5** PrxII binds to tankyrase and preserves its activity in APC-mutant CRC cells. **a** HT29, SW480, and RKO cells were lysed for immunoprecipitation (IP) with anti-TNKS antibody. The IP complexes and total cell lysates (TCL) were immunoblotted for indicated proteins. Total cell lysates were also loaded in parallel with IP samples to identify the positions of PrxI and PrxII proteins (*Arrowhead*). Asterisk indicates non-specific IgG heavy chain. **b** HEK293 cells were co-transfected with Flag-TNKS1 and either control (pQ) or Myc-tagged PrxII-encoding vectors and lysed for IP with anti-M2 (Flag) antibody. Asterisks indicate non-specific IgG heavy chains. **c** An in situ proximity ligation assay shows direct interaction between TNKS and PrxII proteins in the cytosol of SW480 cells. For PrxII-depleted negative control, SW480 cells were transfected with PrxII-1 siRNA. DIC and DAPI-stained images were merged with DuoLink red fluorescence images (Merge). A representative image from two independent experiments is shown. Scale bar, 20 μm. **d** HEK293 cells were co-transfected with truncated mutants of TNKS and Myc-PrxII and lysed for IP with anti-M2 (Flag) antibody. Full-length TNKS1 (1–1327) was expressed as a positive control. The amino acid numbers for each truncate mutant are indicated. **e** PrxII WT and Gly mutants were co-expressed with Flag-TNKS1 in HEK293 cells. The IP complexes were blotted with anti-Myc antibody. **f** HEK293 cells were co-transfected with Flag-TNKS1 and Myc-PrxII vectors as indicated and then treated with 100 μM $H_2O_2$ for 30 min. The Flag-TNKS1 was immunoprecipitated and then subjected to in vitro PARP assay. Data in the graph are means ± s.d. of the $^{32}P$ radioactivity measured in an in vitro assay of PARP activity ($n = 3$, *$P < 0.01$, **$P < 0.005$). The hyperoxidation of 2-Cys Prxs were immunoblotted by anti-Prx-$SO_{2/3}$ antibody. Asterisks marks a non-specific band. **g** SW480 cells were transfected with control or PrxII-1 siRNA for 36 h and then infected with retroviruses expressing siRNA-resistant PrxII WT and G116V mutant. SW480 cells were then grown for 10 days, and the colonies were stained with crystal violet and counted. Data in the graph are the average number of colonies ± s.d. ($n = 3$, *$P < 0.001$). Immunoblots (IB) or $^{32}P$-autoradiographs shown are a representative of three independent experiments. Total cell lysates (TCL) were immunoblotted as control for the amount of indicated proteins. The molecular weight markers are labeled in kilodaltons

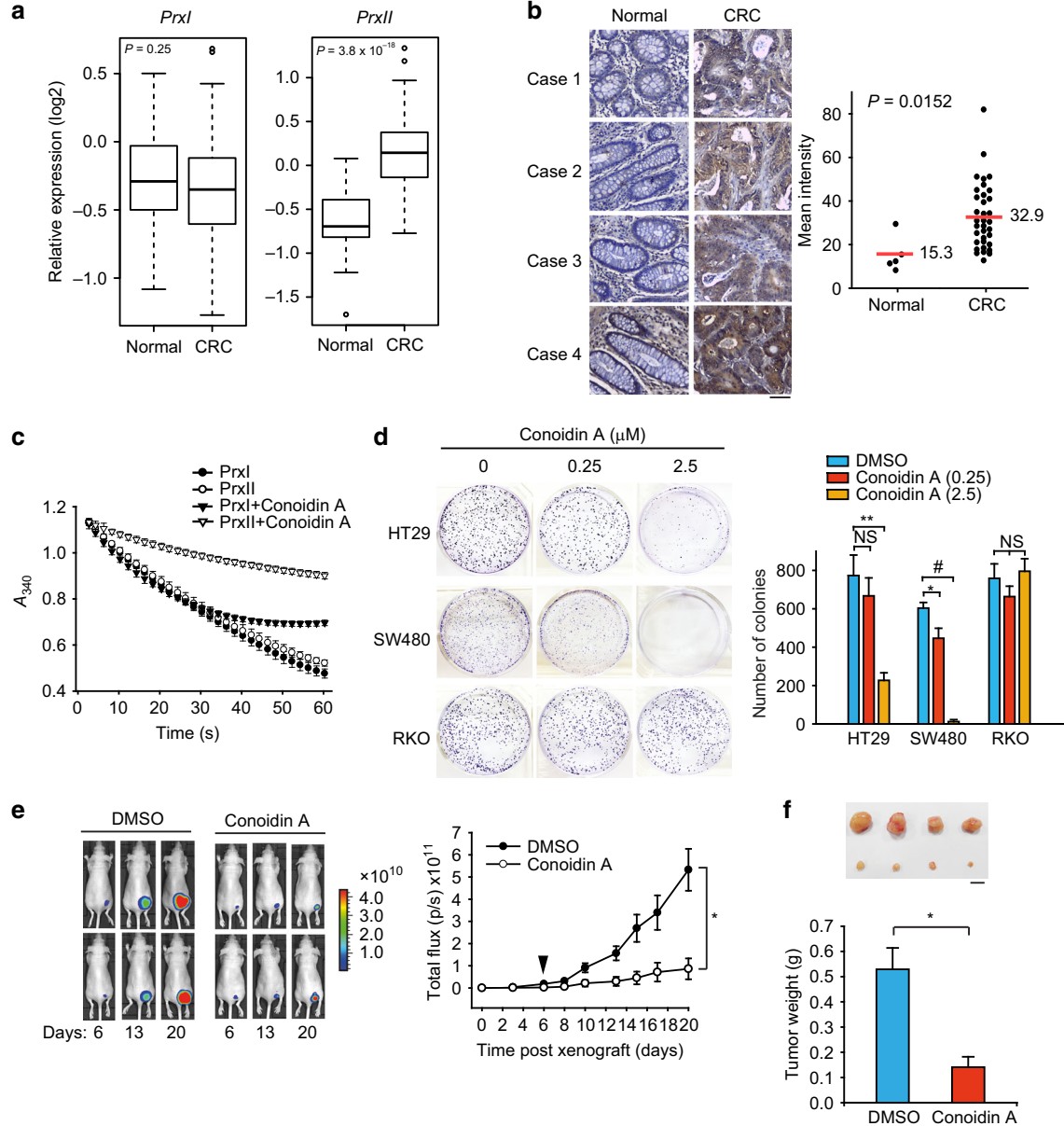

**Fig. 6** PrxII inhibition blocks the tumorigenic growth of APC-mutant CRC cells xenografted to mice. **a** Expression levels of PrxI and PrxII genes in healthy individuals (Normal, $n = 26$) and patients with colorectal cancer. (CRC, $n = 155$) from the TCGA COAD database. Box plot represents the average expression of PrxI and PrxII ± s.d. The horizontal line indicates the median value. **b** PrxII immunostaining in the colon tissue array from healthy individuals (Normal, $n = 5$) and patients with colon cancer (CRC, $n = 45$). The IHC images were quantified using the HistoFAXS Tissue Analysis System. Data in the graph are mean intensities ± s.d. The average intensities and P value are indicated. Scale bar, 50 μm. **c** Purified recombinant PrxI and PrxII enzymes were pre-incubated with conoidin A (10 μM) in the reaction mixture for 5 min before $H_2O_2$ addition. The NADPH reduction was monitored by light absorption at 340 nm. Reaction curves shown are means ± s.d. of three independent experiments. **d** HT29, SW480, and RKO cells were grown in the presence of control vehicle or conoidin A for 10 days, and the colonies were stained with crystal violet and counted. Data in the graph are the average number of colonies ± s.d. ($n = 3$, *$P < 0.02$, **$P < 0.01$, #$P < 0.0001$). NS, statistically not significant. **e** HT29-luc2 cells were injected to athymic nu/nu mice and grown for 6 days before first i.p. administration of conoidin A (*Arrowhead*). Conoidin A was administered every 3 days for 28 days (1 mg conoidin A per kg body weight per injection). The bioluminescent signals were taken as described in the Methods. Data in the graph are the mean photon numbers per second ± s.d. ($n = 7$ mice per group, *$P < 0.0001$ with repeated-measured ANOVA). A representative image is shown. **f** After 28 days, the HT29 Luc2-driven tumor tissues were removed, weighed and photographed. Graph shows the average mass ± s.d. of the isolated tumors ($n = 7$ mice per group, *$P < 0.005$). Scale bar, 10 mm

except the acetylated PrxII was barely detected only in SW480 cells. Thus, an unknown PrxII modification accomplishing TNKS interaction awaits further investigation.

In addition to the key signaling function of PrxII as a TNKS-binding partner in APC-mutant CRC, we acknowledge that a general antioxidant function of PrxII somewhat contributes to the survival of CRC cells. In fact, blocking PrxII activity has been shown to activate intrinsic programmed cell death pathway in cancer cells[64], which is ultimately beneficial for eradicating CRC cells. We also expect an additional anti-angiogenic effect of PrxII inhibition in CRC tissues because PrxII has been shown to be essential for the VEGFR2 function in vascular endothelial cells[15].

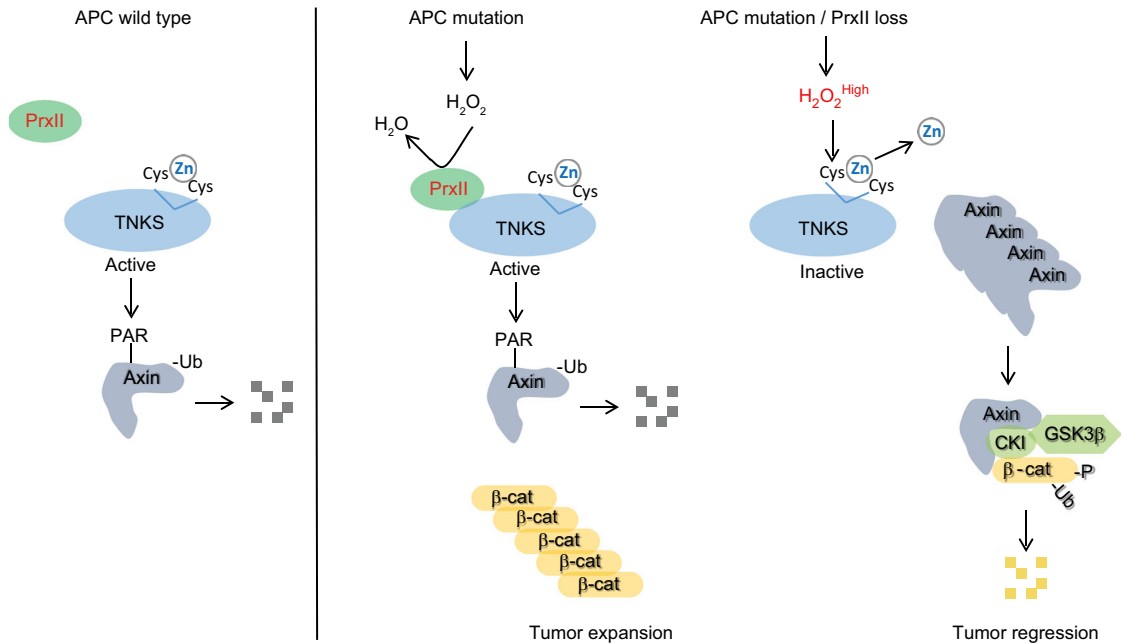

**Fig. 7** PrxII is essential for full TNKS activity that maintains oncogenic β-catenin signaling in APC-mutant CRC cells. The *APC* mutation increases intracellular $H_2O_2$ level and induces PrxII binding to TNKS. PrxII eliminates $H_2O_2$ adjacent to TNKS and sufficiently protects the zinc-binding motif, which is essential for TNKS PARP activity, from oxidative inactivation. However, PrxII loss or inhibition augments the intracellular $H_2O_2$ level, which in turn attacks the Cys residues coordinating a zinc ion and inactivates TNKS. TNKS inactivation leads to the Axin stabilization and consequently β-catenin degradation. Therefore, PrxII loss selectively inhibits the growth of APC mutant CRC cells

Indeed, we confirmed the reduced VEGFR2 kinase activity and angiogenic vessel formation in the polyps from $APC^{Min/+}$; $PrxII^{-/-}$ mice compared to those from $APC^{Min/+};PrxII^{+/+}$ mice. Overall, our study demonstrates that targeting of PrxII may exert specific and broad therapeutic potentials for treating familial adenomatous polyposis as well as *APC*-mutation-positive CRCs.

## Methods

**Materials**. All chemicals, as well as the anti-tubulin (mouse monoclonal, B-5-1-2, 1:8000, T5168) and anti-FLAG antibodies (mouse monoclonal, M2, 1:1000, F3165) were purchased from Sigma-Aldrich. Antibodies against pS33/37pT41-β-catenin (rabbit polyclonal, 1:1000, 9561), β-catenin (rabbit monoclonal, 6B3, 1:1000, 9582), Axin1 (rabbit monoclonal, C76H11, 1:1000, 2087), Axin2 (rabbit monoclonal, 76G6, 1:1000, 2151), GSK3β (rabbit monoclonal, 27C10, 1:1000, 9315), β-actin (rabbit monoclonal, 13E5, 1:1000, 4970), and cyclin D1 (rabbit polyclonal, 1:1000, 2922) were purchased from Cell Signaling Technology. Antibodies against c-Myc (rabbit polyclonal, 1:1000, sc-788), ubiquitin (mouse monoclonal, P4D1, 1:1000, sc-8017), pY279/216-GSK3β (rabbit polyclonal, 1:1000, sc-135653), and tankyrase-1/2 (rabbit polyclonal, 1:1000, H-350) were purchased from Santa Cruz Biotechnology. Anti-active β-catenin (mouse monoclonal, 8E7, 1:1000, 05-665) and anti-Myc tag (mouse monoclonal, 9E10, 1:1000, 05-419), and anti-APC (mouse monoclonal, FE9, 1:1000, MABC202) antibodies were purchased from Millipore. Alexa Fluor 568-conjugated donkey anti-rabbit IgG (1:200, A-21206) and anti-β-TrCP (mouse monoclonal, 1B1D2, 1:1000, 37-3400) antibodies were purchased from Invitrogen. The anti-PAR antibody (rabbit polyclonal, 1:2000, 551813) detecting the poly (ADP-ribose) chain was purchased from BD Bioscience. The anti-Ki-67 antibody (rabbit monoclonal, SP6, 1:200, MA5-14520) was from Thermo Fisher Scientific. Rabbit polyclonal antibodies against PrxI (1:3000), PrxII (1:3000), and Prx-$SO_{2/3}$ (1:1000) were generated as previously described[16, 65]. The rabbit anti-PrxII antisera were affinity purified with recombinant PrxII protein-conjugated agarose gel beads and used for immunofluorescence and PLAs. Wnt3a was purchased from R&D Biosystems. DuoLink in situ fluorescence reagents were purchased from Sigma-Aldrich. The TissueFocus Colon Cancer Tissue Microarray was purchased from OriGene Technologies (Rockville, USA).

siRNA for human PrxI and PrxII were previously described[15]: 5′-ACUCAACU GCCAAGUGAUUUU-3′ (PrxI-1) and 5′-CCACGGAGAUCAUUGCUUUUU-3′ (PrxI-2) for human PrxI; 5′-CGCUUGUCUGAGGAUUACGUU-3′ (PrxII-1) and 5′-AGGAAUAUUUCUCCAAACAUU-3′ (PrxII-2) for human PrxII.

The verified siRNAs for human Axin1 (5′-GGGCAUAUCUGGAUACCUG UU-3′) and Axin2 (5′-GAGUAGCCAAAGCGAUCUAUU-3′) were synthesized as previously described[32].

The verified siRNA mix for human APC (5′-CUGGUAUUUGAGGUGAGAU UU-3′, 5′-GUGCAUAGAGAUAGCUACAUU-3′, and 5′-CAAAUCGAGUGGG UUCUAAUU-3′) were synthesized (Bioneer, Seoul, Korea).

**TCGA analysis**. The expression of *PrxI* and *PrxII* in colon cancer and normal colon tissue samples was measured using microarray data from The Cancer Genome Atlas (TCGA) project (https://tcga-data.nci.nih.gov). Briefly, mRNA expression data were generated by using the Agilent G4502A microarray platform, after which they were processed and normalized as described previously[1]. Gene expression data from 155 colon cancer tissue samples and 26 normal colon tissue samples were included for analysis.

**Cell culture**. All CRC and HEK293 cells were purchased from American Type Culture Collection (Manassas, VA, USA). SW480, DLD1, CoLo205, Colo741, and SW620 cells were cultured and passaged in RPMI 1840 Medium supplemented with 10% fetal bovine serum. HEK293 and RKO cells were cultured in Dulbecco's Modified Eagle's Medium supplemented with 10% fetal bovine serum. HT29 cells were cultured in McCoy's 5A medium supplemented with 10% fetal bovine serum. Mycoplasma contamination was periodically tested in the cell culture supernatants using mycoplasma detection kit (Biotool, USA).

**Immunoblotting and immunoprecipitation**. The intestinal lumen was cleaned by flushing it with ice-cold phosphate-buffered saline (PBS) using a syringe with a blunted needle and incised longitudinally. Intestinal segments without polyps were excised for immunoblotting analysis along with polyps. Tissues were homogenized using a Dounce homogenizer in HEPES-buffered saline containing 10% glycerol, 1 mM EDTA, 2 mM EGTA, 1 mM DTT, 5 mM $Na_3VO_4$, 5 mM NaF, 1 mM AEBSF, aprotinin (5 μg/ml), and leupeptin (5 μg/ml). The cultured cells were rinsed once with ice-cold PBS and then lysed in lysis buffer containing 20 mM HEPES (pH 7.0), 1% Triton X-100, 150 mM NaCl, 10% glycerol, 1 mM EDTA, 2 mM EGTA, 1 mM DTT, 5 mM $Na_3VO_4$, 5 mM NaF, 1 mM AEBSF, aprotinin (5 μg/ml), and leupeptin (5 μg/ml). Tissue homogenates and cell lysates were centrifuged at $15,000 \times g$ for 15 min, and the protein concentrations were determined by Bradford assay (Pierce). Protein samples were mixed with SDS sample buffer and boiled for 5 min. The proteins were separated by SDS-PAGE and then transferred onto nitrocellulose membranes by electroblotting for 1 h. The membranes were blocked with 5% bovine serum albumin (BSA) or 5% dry skimmed milk in Tris-buffered saline containing 0.05% (v/v) Tween-20 (TBST) for 2 h and then incubated with the appropriate primary antibody in blocking buffer for 2 h at room temperature. After washing three times with TBST, membranes were incubated with horseradish peroxidase-conjugated secondary antibody (Amersham Biosciences) in blocking

buffer. The immune-reactive bands were detected with an enhanced chemilumi-nescence kit (AbFrontier, Korea) and quantified by a LAS-3000 imaging system (Fuji Film, Japan). When necessary, the membranes were stripped by shaking them for 60 min at 37 °C in 67 mM Tris (pH 6.7), 2% SDS, and 100 mM β-mercap-toethanol and then reprobed with the appropriate pan-antibody. For IP, the clarified cell lysates (0.5 ~ 1 mg protein) were pre-cleared with 30 µl of protein-A/G Sepharose 4 Fast Flow beads (Amersham Biosciences) for 1 h. The supernatant was incubated overnight with 3 µg of the appropriate antibody with rotation and then precipitated by mixing with 30 µl of protein-A/G beads at 4 °C for an additional 3 h. The beads were washed three times with 1 ml of lysis buffer and then subjected to an in vitro PARP assay or to immunoblotting.

**β-catenin/TCF transcription reporter assay.** SW480 cells were seeded on 12-well plates and transfected with the pTOPflash or pFOPflash plasmid. To normalize transfection efficiency, cells were cotransfected with pRL-TK Renilla Luciferase control plasmid. After transfection, cells were incubated in complete medium for 24 h and then lysed with reporter lysis buffer. Luciferase activity was measured in triplicate by Dual-Luciferase reporter assay (Promega). Data were reported as fold induction compared with control siRNA group after normalization of transfection efficiency.

**RNA-Seq analysis.** Four samples of the siRNA-transfected HT29 cells were pre-pared for high-throughput mRNA sequencing. Total RNA of 1 µg was extracted from each sample and the mRNA library (insert size of ~300 bp) was created using the TruSeq RNA Library Preparation kit v2 (Illumina). Paired-end transcriptome sequencing (101 bp read length) was performed on Illumina HiSeq 2500. The number of reads for each sample ranged from 69.4 million to 74.8 million. The raw sequencing data were deposited in the GEO database (Accession Number: GSE81429). RNA-Seq data were aligned to the human genome (hg19 from UCSC) using MapSplice v2.1.7 after standard quality check and trimming with FastQC and Fastx-toolkit, respectively. The mapping rate of reads was between 96.5 and 96.8%. RSEM v1.2.12 was used to estimate the transcriptome abundance for refGene mRNAs. Differentially expressed genes were identified using DESeq2 with the FDR cutoff of 0.05.

**In vitro PARP assay.** The immunocomplex-bound beads were incubated in 40 µl of an assay buffer (50 mM Tris-HCl pH 8.0, 4 mM MgCl$_2$) containing 4 µCi of γ-$^{32}$[P]-NAD$^+$ at 25 °C for 30 min[66]. The reaction was stopped by the addition of 2 × SDS sample buffer. The samples were boiled and then separated on SDS denaturing gel. The gel was vacuum-dried and autoradiographed with an imaging plate. The radioactivity recorded on the plate was read and quantified by a Fujifilm Bio-imaging Analyzer System (BAS)-3000.

**Plasmid construction and site-directed mutagenesis.** A plasmid containing a full-length complementary DNA of human tankyrase-1 was purchased from Open Biosystems (mRNA accession number BC098394). The full-length sequence of tankyrase-1 was PCR-amplified using the forward and reverse primers (5′-ATAAG AATGCGGCCGCGGCGGCGTCGCGTCGCTC-3′ and 5′-GAAGATCTCTAGGT CTTCTGCTCTG-3′, where the NotI and BglII sites are underlined, respectively) and inserted into the p3 × FLAG CMV9 vector to generate FLAG-tagged TNKS1. For the domain-mapping experiment, the various tankyrase-1 fragments were PCR-amplified and subcloned into the p3 × FLAG CMV9 vector using the following forward and reverse primer sets: for amino acid residues 1–158, 5′-ATA AGAATGCGGCCGCGGCGGCGTCGCGTCGCTC-3′ and 5′-GAAGATCTCTA GGCCGCCTCGGGGCTCTC-3′; for amino acid residues 158–595, 5′-ATAAGA ATGCGGCCGCCGGAGTTAGCAGCACAGCAC-3′ and 5′-GAAGATCTCTACA AAGCAGTCTGACCAAGGG-3′; for amino acid residues 596–1022, 5′-ATAAGA ATGCGGCCGCGCATAGAGCCGCCCTAGCAGG-3′ and 5′-GAAGATCTCTAT CCTTCCTTCCTTTCTGTTCC-3′; for amino acid residues 1023–1327, 5′-ATAA GAATGCGGCCGCAGAAGTTGCTGGTCTTGAC-3′ and 5′-GAAGATCTCTA GGTCTTCTGCTCTG-3′, where the NotI and BglII sites are underlined, respec-tively. The Escherichia coli expression plasmid for GST-TNKS1 (1023–1327) was a kind gift from Chang-Woo Lee (Sungkyunkwan University School of Medicine)[67]. The retroviral vectors (pQ-CXIX) expressing the Myc-tagged siRNA-resistant PrxII WT, C172S single mutant, and C51/172 S double mutant were prepared as pre-viously described[15].

Site-directed mutagenesis for amino acid substitution was performed using the QuikChange kit (Stratagene). The double-stranded primers for Cys-to-Ser substitutions on tankyrase-1 were as follows: for C1163S mutant, (sense) 5′-GTT GAGGGAGCGGTTCTCCCACCGACAGAAGGAAG-3′; for C1234S mutant, (sense) 5′-GGAGGAGGAACAGGCTCCCCTACACACAAGGAC-3′; for C1242S mutant, (sense) 5′-CACAAGGACAGGTCATCCTATATATATGTCACAGAC-3′; for C1245S mutant, (sense) 5′-CAGGTCATGCTATATATCTCACAGACAAATGC TCTTC-3′; for C1252Smutant, (sense) 5′-GACAAATGCTCTTCTCTAGAGTGA CCCTTGGG-3′.

The double-stranded primers for Gly-to-Val substitutions on human PrxII were as follows: (sense) 5′-GCGCGCATCGTAAAGCCAGCCCCTG-3′ for the G9V mutant; (sense) 5′-GCGGTGGTTGATGTCGCCTTCAAAG-3′ for the G23V mutant; (sense) 5′-CTGAGGGATTACGTCGTGCTGAAAAC-3′ for G116V mutant.

Retroviral pQ vector expressing β-catenin S37A mutant was prepared by PCR subcloning from pBI-EGFP-β-catenin(S37A) construct described previously[68]. All constructs and mutations were verified by nucleotide sequencing.

**Zinc determination.** Zinc ions were monitored in an aqueous solution using 4-(2-pyridylazo) resorcinol (PAR)[38]. The glutathione S-transferase (GST)-fused TNKS1 PARP (1023–1327) proteins were expressed in E. coli grown in LB medium supplemented with 100 µM ZnCl$_2$ and purified by affinity chromatography using Glutathione Sepharose 4B Fast Flow beads according to the manufacturer's pro-tocol (GE Healthcare Life Sciences). The purity of GST-TNKS1 PARP proteins (>99.5%) was verified by densitometry and then extensively dialyzed in Chelex100-treated buffer containing 25 mM HEPES (pH 7.0) and 2 mM DTT to eliminate unbound zinc ions. The GST-TANK1 PARP proteins were incubated with 500 µM H$_2$O$_2$ at 30 °C in 200 µl of 40 mM HEPES (pH 7.0) reaction buffer containing 0.1 mM PAR. The formation of the PAR$_2$-Zn$^{2+}$ complex was monitored spectro-photometrically at 500 nm with an UV/VIS spectrophotometer (Agilent). The total zinc content in the purified GST-TNKS1 PARP proteins used in the assay was determined by addition of 0.5 mM p-chloromercuribenzoic acid to the reaction mixture.

**Peroxiredoxin assay.** A standard peroxidase assay was carried out in a 200 µl reaction mixture containing 250 µM NADPH, 1.5 µM yeast TR, 3 µM yeast Trx, recombinant human Prx (PrxI, 4.6 µM; PrxII, 16.4 µM), and 200 µM H$_2$O$_2$ in 50 mM HEPES (pH 7.0) containing 1 mM EDTA[69]. The mixture (minus H$_2$O$_2$) was pre-incubated for 5 min in the presence or absence of conoidin A (100 µM), and then the reaction was initiated by adding H$_2$O$_2$. The NADPH oxidation was monitored for 5 min at 30 °C by following absorbance reduction at 340 nm in an Agilent UV8453 spectrophotometer (Hewlett Packard, USA). The initial rate of reaction was calculated using the linear portion of the curve and expressed as the amount of NADPH oxidized per minute.

**Mouse models.** PrxI$^{+/-}$ and PrxII$^{+/-}$ C57BL/6 mice[15] were crossed to APC$^{Min/+}$ mice on a C57BL/6 background (Jackson Laboratory, Bar Harbor, USA). The mice were bred and maintained in our specific pathogen-free mouse facility to establish the compound mutant littermates: APC$^{Min/+}$/PrxI$^{+/+}$, APC$^{Min/+}$/PrxI$^{+/-}$, APC$^{Min/+}$/PrxI$^{-/-}$, APC$^{Min/+}$/PrxII$^{+/+}$, APC$^{Min/+}$/PrxII$^{+/-}$, and APC$^{Min/+}$/PrxII$^{-/-}$ mice. The littermates were genotyped by genomic PCR of mouse tail DNA with specific primers: APC$^{Min}$ (5′-GCCATCCCTTCACGTTAG-3′ for WT; 5′-TTCTGAGAAAGACAGAAGTTA-3′ for mutant; 5′-TTCCACTTTGGCATA AGGC-3′ for common), PrxI (forward, 5′-CTGGAAACCTGGCAGTGATA-3′; reverse, 5′-CTGTGACTGATAGAAGATTGGT-3′), PrxII (forward, 5′-GAT GATCTCCGTGGGGCAAACAAA-3′; reverse, 5′-ATGGCCTCCGGCAACGCGC AAATC-3′), Neo cassette (forward, 5′-GCTTGGGTGGAGAGGCTATTCG-3′; reverse, 5′-GTAAAGCACGAGGAAGCGGTCAGC-3′). All mouse experiments were approved by the Institutional Animal Care and Use Committee (IACUC) of Ewha Womans University, South Korea, and conformed to the ARRIVE guide-lines[70]. The animal study was double-blinded by separating animal breeding and tissue analysis. The animal experiments were repeated twice, and the numbers of animals in the experimental groups are described in the figure legends.

For a tumor xenograft model, the mice were anesthetized by inhalation of isoflurane gas (N$_2$O:O$_2$/70%:30%) and then subcutaneously injected with HT29-luc2 cells (2.5 × 10$^5$ cells) suspended in 200 µl of PBS. Intraperitoneal administration of conoidin A compound (286 µM in DMSO) was begun 6 days after cell injection and repeated every 3 days. The bioluminescent imaging was carried out with IVIS Lumina Series III (Perkin Elmer). For each imaging session, the luciferin in PBS (a total of 150 mg Luciferin per kg body weight) was administered into the peritoneal cavity according to manufacturer's protocol. Up to four animals were maintained in the instrument with the integral anesthetic manifold and imaged 10 min after Luciferin injection. IVIS imaging system acquires a photographic image of the mice and a quantitative bioluminescent signal, which are then overlaid by each other.

**Histology immunohistochemistry and immunofluorescence staining.** The 12-week-old male mice were anesthetized by inhalation of isoflurane gas (N$_2$O:O$_2$/70%:30%) before transcardiac perfusion-fixation with heparinized saline containing 3.7% formaldehyde. The intestines were then excised and cut into two segments, small intestine and colon, both of which were opened longitudinally and rolled outwards with the mucosa. The rolled intestines were paraffin-embedded and sectioned by rotary microtome (Leica RM2255). Three serial tissue sections (10 µm in thickness) were stained with hematoxylin and eosin. The rolled intestines were immediately embedded in OCT medium and frozen on dry ice. A cryotome was used to cut the tissue into 10 µm sections, which were placed on Superfrost Plus slides (Surgipath Medical Inc., UK), dried at room temperature, and kept at −80 °C until they were thawed for immunostaining.

For immunohistochemistry, the paraffin sections were de-waxed in xylene and rehydrated in ethanol. Antigen retrieval was subsequently performed by boiling the sections in a sodium citrate buffer (pH 6.0). Tissue sections were then incubated with anti-Ki-67 antibody (1:200 dilution) and the affinity-purified anti-PrxII antibody (1:500 dilution) for 48 h at 4 °C. After washing three times with PBS,

sections were incubated with a peroxidase-conjugated secondary antibody and stained with 3′,3′-diaminobenzidine (DAB) substrate solution. The nuclei were further stained with hematoxylin. The DAB-stained images were obtained and quantified using the HistoFAXS Tissue Analysis System (TissueGnostics, USA).

For immunofluorescence staining, the paraffin or frozen tissue sections were blocked with 5% normal rabbit serum (Vector Laboratories) in PBST (0.3% Triton X-100 in PBS) for 1 h at room temperature. Sections were then incubated overnight at 4 °C with primary antibody (1:500 dilution for anti-PrxII antibody; 1:100 dilution for anti-BrdU antibody). After several PBST washings, the samples were incubated for 2 h at room temperature with Alexa Fluor 568-conjugated donkey anti-rabbit IgG antibody. Sections were counterstained with 4′,6′-diamidino-2-phenylindole (DAPI, Sigma-Aldrich) for 30 min and mounted using Vectashield mounting medium. The fluorescence images were obtained in three random fields per tissue section at 100× magnification using a LSM 510 Meta confocal microscope equipped with argon and helium-neon lasers (Carl Zeiss, Germany).

**Quantitative real-time PCR**. Total RNA was isolated from mouse intestinal polyps using the RNeasy kit (Qiagen) and used for cDNA synthesis with Superscript first strand cDNA synthesis kit (Invitrogen). The quantitative real-time PCR (qPCR) was performed in triplicate using SYBR Green qPCR Master Mix on a fluorescent temperature cycler (ABI Prism 7300 sequence detection system, Applied Biosystems). The fluorescence signals were quantified by a comparative cycle threshold method and normalized with β-actin mRNA and 18S rRNA levels as endogenous references. The qPCR primers are as follows: for mouse β-catenin, (sense) 5′-AC TGCTGGGACTCTG-3′ and (antisense) 5′-TGATGGCGTAGAACAG-3′; for mouse c-Myc, (sense) 5′-GCTGCTTAGACGCTGGATTT-3′ and (antisense) 5′-CACCGAGTCGTAGTCGAGGT-3′; for mouse Axin-1, (sense) 5′-ACGGTACA ACGAAGCAGAGAGCT-3′ and (antisense) 5′-CGGATCTCCTTTGGCATTCG GTAA-3′; for mouse cyclin D1, (sense) 5′-GCAAGCATGCACAGACCTT-3′ and (antisense) 5′-GTTGTGCGGTAGCAGGAGA-3′; for mouse β-actin, (sense) 5′-TGGATCAGCAAGCAGGAGTATG-3′ and (antisense) 5′-GCATTTGCGGTG GCAGAT3′; for 18S ribosomal RNA, (sense) 5′-GGCCGTTCTTAGTTGGTGGA GCG-3′ and (antisense) 5′-CTGAACGCCACTTGTCCCTC-3′.

**Statistical analysis**. Unless otherwise stated, data were analyzed with either the Student's t-test for comparisons between two groups or ANOVA with Tukey's "honestly significant difference" post hoc test for multiple groups (SPSS 12.0K for Windows, SPSS, Chicago, IL, USA) to determine the statistical significance (P value). A $P < 0.05$ was considered to be statistically significant.

**Data availability**. The RNA sequencing data that support the findings of this study have been deposited in the GEO database (Accession Number: GSE81429). The data that support the findings of this study are available from the corresponding author upon reasonable request. The authors declare that all the data supporting the findings of this study are available within the paper and its supplementary information files.

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

## Acknowledgements

We thank Drs. Chang-Woo Lee and Goo Taeg Oh for sharing plasmids and knockout mice, respectively, and Sue Goo Rhee for valuable comments. This study was supported by grants from the National Research Foundation of Korea (2014R1A2A1A01006934) and from the National R&D Program for Cancer Control (1420280). This study was also supported in part by grants from the National Research Foundation of Korea (2012M3A9C5048709 and 2012R1A5A1048236). D.H.K. was a recipient of a Basic Science Research Program Award from the National Research Foundation of Korea (NRF-2014R1A6A3A04058006). E.-H.J. was supported by the 2016 sabbatical year research grant of the University of Seoul.

## Author contributions

D.H.K., D.J.L., S.L., S.-Y.L., Ye.K., and Yo.K.: Performed experiments; D.H.K., D.J.L., D.-K.L., E.-H.J., and S.W.K.: Designed experiments and interpreted data; Y.J., S.L., and J.-S.L.: Performed bioinformatics analysis; D.-Y.Y.: Provided knockout mice; S.W.K.: Conceived the project, supervised the study, and wrote the manuscript.

## Additional information

**Competing interests:** The authors declare no competing financial interests.

