## [Peer Review file · Nature Communications]

Reviewers' comments:

Reviewer #1 (Remarks to the Author):

In this manuscript, authors describe a new mechanism that regulates tankyrase activity by PrxII in APC deficient cells. This is a very nice study despite of lacking of mechanistic understanding of APC mutant selectivity. Overall, data supporting the proposed mechanism are convincing.

Major point:

1. It is still not clear whether growth inhibitory activity of PrxII depletion is mediated by tankyrase or beta-catenin inhibition. Depletion of PrxII inhibited the proliferation of SW480 cells (Fig. 3e). However, it has been shown previously that although TNKS clearly decreases the level of total beta-catenin in SW480 cells, it has a minimal effect on beta-catenin signaling (PMID: 24419084). The argument is that nuclear beta-catenin is protected from degradation by LEF1 and BCL9L in this cell line. Authors should check the effect of PrxII siRNA and tankyrase inhibitor on the expression of beta-catenin target genes in SW480 cells and examine whether tankyrase inhibitor can inhibit the proliferation of SW480 cells. Beta-catenin S37A rescue experiment will certainly strengthen their argument.

2. Fig. 1e. Authors should examine Axin1 mRNA level to make sure the mRNA level of Axin1 is not elevated in PrxII^{-/-} sample.

Reviewer #2 (Remarks to the Author):

Summary

In this paper Kang and colleagues show that in human colorectal cancer cells with APC mutation, the antioxidant peroxiredoxin 2 (Prx2) binds to and protects the poly(ADP-ribose) polymerase (PARP) tankyrase from oxidative inactivation by H₂O₂, preventing the release of zinc-binding motif-bound zinc, thereby allowing degradation of the scaffold protein Axin1, which is part of the b-catenin destruction box that assists b-catenin degradation.

Knockout of PRX2 in mice with inactivating mutation of APC reduced the frequency and size of polyps, and significantly promoted mice survival. In polyps from these animals the levels of b-catenin were decreased, whereas those of Axin1, were increased. These effects were specific to Prx2, since they were not seen after KO of PRX1 in APC/min/+.

In human colorectal cancer cells, Prx2 depletion inactivated tankyrase, which prevented Axin1 degradation, thereby reducing the level of its target b-catenin and target genes. Tankyrase was shown to specifically bind Prx2 through the ARC4-5 repeats and a glycine-containing tankyrase-binding motif present in Prx2 but not in Prx1. An inhibitor of Prx2 inhibited expansion of APC-mutant colon cancer cells in vitro and in an vivo xenograft.

General comments

This study proposes a surprising pro-oncogenic function of the antioxidant Prx2 in the context of colorectal cancer with APC mutation, whereby by binding to tankyrase, Prx2 prevents the oxidative inactivation of this enzyme, thereby preventing Axin1-dependent degradation of b-catenin. It is supposed that Prx2 scavenges the H₂O₂ locally in the vicinity of tankyrase, thereby preventing its oxidative inactivation. Surprisingly, these findings are only observed in cells with APC mutations for reasons that are not rationalized in this study. In conclusion, the data provided provide molecular and in vivo data that constitute a strong case for an APC-, Axin1- and b-catenin-dependent oncogenic role of Prx2 in colorectal cancer. Data provided appear solid and well controlled and clearly support the authors' conclusions. Ironically, the molecular mechanism of the oncogenic propriety of Prx2 proposed here is built on our ignorance in the actual function of both APC and Axin1 in colon carcinogenesis. The study provides a very strong case for the specificity of this mechanism towards colon cancer with APC mutation without giving a hint of the nature of the

mechanism responsible for this specificity. Overall the study is very interesting and novel. It is surprising that such a mechanism only applies to colorectal cells with a APC mutation, which shows that cancer evolves its own mechanisms to succeed.

Specific comments

1. Figure 2d nicely shows that reintroducing WT Prx2 in cells with a Prx2 Kd restores the levels of b-catenin, whereas a mutant lacking both catalytic Cys residues does not: what is the effect of the single Cys172S mutant?
2. Is there a cumulative effect of the PRX1-Prx2 double Kd on b-catenin/Axin1 levels?
3. Fig 4: exogenous H₂O₂ inhibits Tankyrase PARP activity in both RKO and HT29, whereas depletion of Prx2 only works in HT29 to degrade b-catenin. To clearly show the cause-and effect relationship between H₂O₂ scavenging at the vicinity of tankyrase and loss of this activity to perform a H₂O₂ titration experiments comparing RKO and RKO with APC Kd?
4. Fig 4A: There is no increase of TANKS levels seen upon depletion of Prx2 in HT29 cells: explain?
5. 4F: why is the migration of TANKS Cys Mutants faster than the other forms of the protein?
6. Fig5: explain what are the data indicating direct interaction between TANKS and Prx2?
7. Fig5F: explain why in HEK293 cells, endogenous Prx2 is not able to prevent H₂O₂ from inactivating TANKS?
8. 5F: why only the endogenous Prx is sensitive to hyperoxydation? In page 11, the sentence Hence, MyC-Prx2 prevented the hyperoxydation of endogenous enzyme... is unclear: it should be rephrased and explained better?
9. Fig5: is it possible to convert Prx1 into Prx2 with regards to its effect on TANKS by inserting the Arg residue the TANKS-ARC binding consensus motif present in Prx2?
10. Fig6C: why is there a difference in the effect of conoidin A on Prx1 and Prx2? This is surprising as it would indicate that the chemical works differently on the two isoenzymes.
11. Page 9: what is the purpose of the blunt statement that exogenous H₂O₂ led to the hyperoxydation of endogenous 2-Cys Prxs?
12. Page 15: why is the Prx2-TANKS interaction only seen in APC mutant is poorly addressed in the text: what could be the posttranslational modification specifically linked to the APC mutation?

Responses to Reviewer #1:

In this manuscript, authors describe a new mechanism that regulates tankyrase activity by PrxII in APC deficient cells. This is a very nice study despite of lacking of mechanistic understanding of APC mutant selectivity. Overall, data supporting the proposed mechanism are convincing.

Response: We all authors sincerely thank for carefully reading our manuscript and providing valuable comments and critics. We performed a number of additional experiments and prepared the answers to every comment. Now, we indicate all changes with blue color in the revised manuscript.

Major point:

1. It is still not clear whether growth inhibitory activity of PrxII depletion is mediated by tankyrase or beta-catenin inhibition. Depletion of PrxII inhibited the proliferation of SW480 cells (Fig. 3e). However, it has been shown previously that although TNKS clearly decreases the level of total beta-catenin in SW480 cells, it has a minimal effect on beta-catenin signaling (PMID: 24419084). The argument is that nuclear beta-catenin is protected from degradation by LEF1 and BCL9L in this cell line. Authors should check the effect of PrxII siRNA and tankyrase inhibitor on the expression of beta-catenin target genes in SW480 cells and examine whether tankyrase inhibitor can inhibit the proliferation of SW480 cells. Beta-catenin S37A rescue experiment will certainly strengthen their argument.

Answer: As the reviewer suggested, we have performed additional experiments to confirm the role of PrxII in SW480 cells.

- ① In previous manuscript, we showed that PrxII depletion downregulated expression of β -catenin target genes in HT29 cells. Thus, we carried out the same mRNA sequencing with SW480 cells depleted of PrxII expression (GEO Series GSE81429). The RNA-Seq analysis clearly showed that PrxII depletion also downregulated expression of β -catenin target genes in SW480 cells (**revised Supplementary Fig. 5c**). Importantly, some genes (CCND1, ID2,

S100A4, and EDN1) were overlapped but the others were distinct between two cell types, which means cell-type specificity of β -catenin transcription. This result is described in the text (page 8, lane 1).

- ② We found the public whole transcriptome data using SW480 cells treated with TNKS inhibitor (PMID: 24747911; GEO ID: GSE55624) and performed GSE analysis of β -catenin target genes. As a result, treatment of TNKS inhibitor (NVP-TNKS656) indeed downregulated expression of at least 11 β -catenin target genes in SW480 cells, among which eight genes were completely matched with those found in PrxII-depleted SW480 cells (see **Figure below**). Furthermore, TNKS inhibitor (XAV939) evidently inhibited proliferation of both HT29 and SW480 cells in a dose-dependent manner (**revised Supplementary Fig. 6b**). It is noted that SW480 cells were less sensitive to TNKS inhibitor than HT29 cells. The result is described in the text (page 9, 5th lane from bottom).

- ③ Retroviruses expressing β -catenin S37A were prepared and infected HT29 and SW480 cells (see **Figure below**). As a result, expression of β -catenin S37A nicely rescued the colony-forming abilities of both CRC cells that had been reduced by PrxII depletion (**revised Fig. 3f**), which suggests that growth inhibitory activity of PrxII depletion is mediated by Axin1-

dependent β -catenin degradation. The result is described in the text (page 9, lane 3).

Overall, our additional data further confirm that growth inhibitory activity of PrxII depletion is absolutely mediated by β -catenin inhibition as a consequence of tankyrase inactivation. As the reviewer predicted, β -catenin S37A fully rescued growth inhibition of both HT29 and SW480 cells by PrxII depletion. Regarding TNKS inhibitor, unlike the previous study (PMID: 24419084), we found that TNKS inhibitor truly reduced proliferation of SW480 cells. It is likely that such discrepancy is due to the differential gene expression profile mediated by β -catenin-associated complex. In fact, the previous paper claims that TNKS inhibitor reduced β -catenin associated with TCF and BCL9 but not with LEF and BCL9L. Therefore, we carefully argue that expression profile of β -catenin target genes may differ depending on β -catenin-associated transcription complex. Our whole-transcriptome analysis using public data brought a subset of β -catenin target genes distinct from β -catenin target genes shown in the previous paper.

2. Fig. 1e. Authors should examine Axin1 mRNA level to make sure the mRNA level of Axin1 is not elevated in PrxII^{-/-} sample.

Answer: This is a good comment. We have examined the mRNA level of Axin1 in PrxII^{+/+} and PrxII^{-/-} polyp samples by quantitative real-time PCR. As a result, we found that the mRNA levels of Axin1 and β -catenin was unaffected by PrxII deletion (**revised Supplementary Fig. 1f**). However, the mRNA levels of other β -catenin target genes, cyclin D1 and c-Myc, were markedly reduced by PrxII deletion. Thus, the results clearly support that Axin1 is regulated by PrxII in a protein level. This

result is described in the text (page 6, lane 1).

Responses to Reviewer #2:

General comments

This study proposes a surprising pro-oncogenic function of the antioxidant Prx2 in the context of colorectal cancer with APC mutation, whereby by binding to tankyrase, Prx2 prevents the oxidative inactivation of this enzyme, thereby preventing Axin1-dependent degradation of b-catenin. It is supposed that Prx2 scavenges the H₂O₂ locally in the vicinity of tankyrase, thereby preventing its oxidative inactivation. Surprisingly, these findings are only observed in cells with APC mutations for reasons that are not rationalized in this study. In conclusion, the data provided provide molecular and in vivo data that constitute a strong case for an APC-, Axin1- and b-catenin-dependent oncogenic role of Prx2 in colorectal cancer. Data provided appear solid and well controlled and clearly support the authors' conclusions. Ironically, the molecular mechanism of the oncogenic propriety of Prx2 proposed here is built on our ignorance in the actual function of both APC and Axin1 in colon carcinogenesis. The study provides a very strong case for the specificity of this mechanism towards colon cancer with APC mutation without giving a hint of the nature of the mechanism responsible for this specificity. Overall the study is very interesting and novel. It is surprising that such a mechanism only applies to colorectal cells with a APC mutation, which shows that cancer evolves its own mechanisms to succeed.

Response: We all authors sincerely thank for carefully reading our manuscript and providing valuable comments and critics. We performed a number of additional experiments and prepared the answers to every comment. Now, we indicate all changes with blue color in the revised manuscript.

Specific comments

1. Figure 2d nicely shows that reintroducing WT Prx2 in cells with a Prx2 Kd restores the levels of b-

catenin, whereas a mutant lacking both catalytic Cys residues does not: what is the effect of the single Cys172S mutant?

Answer: To see the effect of the single C172S mutant, we infected control or PrxII-depleted HT29 cells with retrovirus expressing PrxII-C172S mutant. Western blotting showed that, like C51/172S double mutant, C172S mutant did not rescue level of β -catenin that had been reduced by PrxII depletion (**See below**), again supporting that the peroxidase activity of PrxII is required for maintaining β -catenin level.

[Figure for reviewer] Effect of PrxII-C172S mutant on β -catenin levels. HT29 cells were transfected with control or PrxII-1 siRNA for 24 hr and then infected with retrovirus expressing siRNA-resistant PrxII C172S mutant for additional 24 hr. Empty retrovirus (pQ) was used for control. The exogenously-expressed PrxII is tagged at C-terminus with a Myc epitope.

We appreciate for this good comment and now describe the result in the revised manuscript (**revised Supplementary Fig.3a; page 7, lane 3 in the text**)

2. Is there a cumulative effect of the PRX1-Prx2 double Kd on b-catenin/Axin1 levels?

Answer: As suggested, we performed double knockdown of PrxI and PrxII in both HT29 and SW480 cells. Western blotting showed that the PrxI/II double knockdown reduced β -catenin level and increased Axin1 level just as much as did the PrxII single knockdown (**See below**). The result is consistent with our previous data, where PrxI had no effect on β -catenin level, and therefore indicates that there is no cumulative effect of PrxI and PrxII on Axin1-dependent β -catenin regulation. This result is not included in the revised manuscript due to the redundancy.

[Figure for reviewer] Effect of PrxI/PrxII double knockdown on levels of Axin1 and β -catenin. HT29 and SW480 cells were transfected with control or PrxII-1 siRNA and immunoblotted (IB) for active β -catenin and Axin1.

3. Fig 4: exogenous H₂O₂ inhibits Tankyrase PARP activity in both RKO and HT29, whereas depletion of Prx2 only works in HT29 to degrade b-catenin. To clearly show the cause-and effect relationship between H₂O₂ scavenging at the vicinity of tankyrase and loss of this activity to perform a H₂O₂ titration experiments comparing RKO and RKO with APC Kd?

Answer: We have shown that APC knockdown induced an increase in cellular H₂O₂ level, a significant decrease in Axin1 level, and subsequent increase in β -catenin level in RKO cells (see Figs. 4d and e). In following H₂O₂ titration experiment, we directly measured TNKS activity in RKO and RKO with APC knockdown. As expected, APC knockdown itself reduced the TNKS activity, possibly due to the increased cellular H₂O₂ level, and further accelerated direct inactivation of TNKS induced by increasing H₂O₂ concentration (**revised Fig. 4f**). The result is described in the text (page 9, 3rd lane from bottom).

[Figure for reviewer] H₂O₂-dependent inactivation of TNKS activity in RKO and RKO with APC knockdown.

RKO cells were treated with or without H₂O₂ at the indicated concentrations for 10 min. The immunoprecipitated TNKS was subjected to *in vitro* PARP assay. Total cell lysate (TCL) was immunoblotted (IB) for indicated proteins to check equal loading and knockdown efficiency (APC).

4. Fig 4A: There is no increase of TANKS levels seen upon depletion of Prx2 in HT29 cells: explain?

Answer: We apologize for causing confusion. Level of TNKS was indeed increased by PrxII depletion in HT29 cells, as also shown in Fig. 4c. Therefore, we replaced it with better immunoblot image.

5. 4F: why is the migration of TANKS Cys Mutants faster than the other forms of the protein?

Answer: We did not notice it but greatly appreciate for carefully examining the blots. We re-examined all the blots from repeated experiments and would say with honesty that the mobility of TNKS Cys mutants was absolutely the same as that of WT. For your reference, we attach two blots from different experiments as shown below. If you generously accept, we hope to keep the previous one as the best blot.

6. Fig5: explain what are the data indicating direct interaction between TANKS and Prx2?

Answer: We have performed two experiments to see the direct interaction between TNKS and PrxII. One was co-IP experiments following simultaneous overexpression of epitope-tagged TNKS1 and PrxII (see Fig. 5b). The other experiment was proximity ligation assay (PLA) using Duolink probes in CRC cells (see Fig. 5c and Supplementary Fig. 7). Interaction mapping studies were also based on direct interaction of TNKS and PrxII proteins (see Figs. 5d and e). Hence, all these evidences indicate direct interaction between TNKS and PrxII. To clear confusion, we slightly modified the sentences in the text (page 11, lane 4)

7. Fig5F: explain why in HEK293 cells, endogenous Prx2 is not able to prevent H₂O₂ from inactivating TNKS?

Answer: We previously confirmed that PrxII did not interact with TNKS at endogenous level in HEK293 cells expressing APC wild type (see **Figure #1 below**), just like the case of RKO cells. Moreover, we showed that exogenous H₂O₂ concentration (100 μ M) sufficiently inactivated endogenous 2-Cys Prxs as well as TNKS in HEK293 cells (see lanes 1 and 2 in Fig. 5f) and even in APC-mutant CRC cells (see **Figure #2 below**). Therefore, we mentioned in the text that exogenous H₂O₂ directly inactivates both TNKS and PrxII regardless of their interaction (page 9, lanes 13 - 14). In addition, we have chosen HEK293 cells as suitable non-CRC cells for investigating protein-protein interaction following artificial overexpression of TNKS and PrxII. This notion is described in the text (page 11, lane 5)

[Figure #1 for reviewer] No interaction between TNKS and PrxII in HEK293 cells. HEK293 cells were lysed for immunoprecipitation (IP) with anti-TNKS antibody. The IP complexes and total cell lysate (TCL) were immunoblotted for indicated proteins. Total cell lysates were also loaded in parallel with IP samples to identify the positions of PrxI and PrxII proteins (*Arrowhead*).

[Figure #2 for reviewer] exogenous H₂O₂ concentration (100 μM) sufficiently inactivated endogenous 2-Cys Prxs as well as TNKS in SW480 cells. Originated from experiment in Fig. 4b.

8. 5F: why only the endogenous Prx is sensitive to hyperoxydation? In page 11, the sentence Hence, MyC-Prx2 prevented the hyperoxydation of endogenous enzyme... is unclear: it should be rephrased and explained better?

Answer: We have previously shown that human PrxI and PrxII are inactivated by hyperoxydation during the peroxidase reaction (PMID: 12161445; J Biol Chem. 2002). However, we have recently found that PrxII protein with a Myc tag sequence at carboxyl terminus exhibits a strong peroxidase activity in the reaction mixture without any indication of hyperoxydation (**revised Supplementary Fig. 8c; see below**). Since numerous studies have claimed that the C-terminal modification on PrxI and PrxII protein confers resistance to hyperoxydation (Parmingiani RB et al, PNAS 2008 [PMID: 18606987]; Koo KH et al, ABB 2002 [PMID: 11795888]; Wood ZA et al, Science 2003 [PMID: 12714747]), we speculate that addition of Myc tag to the C-terminus may cause a similar structural change on PrxII, which then converts to hyperoxydation-resistant form.

[Figure for reviewer] Peroxidase activity assay using purified recombinant proteins of PrxII and PrxII with a Myc tag (PrxII-Myc). Typical peroxidase assay was conducted in a reaction mixture containing 50 mM HEPES (pH 7.0), 4.3 μ M human Trx, 0.165 μ M rat liver TrxR, 200 μ M NADPH, 100 μ M H₂O₂, and 3.4 μ M PrxII.

Consistent with the *in vitro* activity shown above, the Myc-tagged PrxII expressed was not hyperoxidized in HEK293 cells treated with exogenous H₂O₂. Therefore, we thought that, due to its strong activity, the expressed Myc-PrxII could prevent the hyperoxidation of endogenous PrxII as well.

Based on comments #7 and #8, we rephrased the sentences in the text (page 11, 6th lane from bottom ~ page 12, lane 3)

9. Fig5: is it possible to convert Prx1 into Prx2 with regards to its effect on TANKS by inserting the Arg residue the TANKS-ARC binding consensus motif present in Prx2?

Answer: This is also good suggestion that we have discussed. To test it, we prepared the PrxI-T111R mutant and performed co-IP experiment with TNKS1. The result, however, showed that a single replacement of Arg residue was insufficient to elicit the interaction of PrxI and TNKS1, which suggests the involvement of more complex molecular interaction between PrxII and TNKS1. Since 3D structure of the hyperoxidized human PrxII protein, which forms decamer, is only available in protein data bank (Schroder E et al, *Structure*. 2000; PMID: 10873855), we cannot simulate the detailed molecular interaction between PrxII and TNKS ARC domains. Yet, we believe that it would be possible if the structure of reduced form of PrxII is solved in the future. This result is described in the revised manuscript (page 14, 4th lane from bottom).

[Figure for reviewer] PrxI-T111R mutant does not interact with TNKS1. Flag-TNKS1 and Myc-PrxI were co-expressed in HEK293 cells and subjected to immunoprecipitation (IP) with anti-Flag antibody. The IP complexes and total cell lysate (TCL) were immunoblotted for indicated proteins. Total cell lysates were also loaded in parallel with IP samples to identify the positions of PrxI and TNKS proteins (*Arrowhead*).

10. Fig6C: why is there a difference in the effect of conoidin A on Prx1 and Prx2? This is surprising as it would indicate that the chemical works differently on the two isoenzymes.

Answer: Our result can properly be explained by biophysical and biochemical evidences in two independent studies (Nguyen JB et al, Chem Biol 2013 [PMID: 23891152]; Haraldsen JD et al, Org & Bimol Chem 2009 [PMID: 21359112]), where conoidin A covalently inactivates PrxI and PrxII in a different mechanism. The first study showed that conoidin A promotes hyperoxidation of PrxI from parasitic nematodes, whereas the other showed that conoidin A inhibits hyperoxidation of PrxII from protozoan parasite. Although Prxs in lower eukaryotes, such as protozoa and nematode, cannot be matched with human orthologs due to high sequence similarity, it is clear that the mode of action of conoidin A is definitely distinct between major 2-Cys Prxs, PrxI and PrxII. We already mentioned this notion with references in the text (page 13, lane 1).

11. Page 9: what is the purpose of the blunt statement that exogenous H₂O₂ led to the hyperoxidation of endogenous 2-Cys Prxs?

Answer: As mentioned in comment #7, we tried to convince that exogenous H₂O₂ concentration (100 μM) sufficiently inactivated endogenous 2-Cys Prxs as well as TNKS in APC-mutant CRC cells. We think that this sentence is no longer necessary because it is described in detail in page 11. So, we

remove this sentence to clear confusion.

12. Page 15: why is the Prx2-TANKS interaction only seen in APC mutant is poorly addressed in the text: what could be the posttranslational modification specifically linked to the APC mutation?

Answer: As we discussed in previous manuscript, numerous studies have shown that PrxII function and activity is regulated by post-translational modification, such as acetylation and phosphorylation. Nonetheless, we failed to see any of known modifications we tested (see Discussion). In the meantime, we have focused on a previous report in which N-terminal acetylation occurs exclusively on PrxII after demethionylation (Seo JH et al. J. Biol. Chem., 2009; PMID: 19286652). [REDACTION]

REVIEWERS' COMMENTS:

Reviewer #1 (Remarks to the Author):

Authors have partially addressed issues that I raised in the last round of review. Although the beta-catenin S37A rescue experiment is good, AXIN2, the most widely studied b-catenin target gene, is not affected by TNKS inhibitor or PRXII siRNA in SW480 cells. Down-regulation of several other genes in SW480 cells might not be mediated by b-catenin inhibition. Authors should minimally comment on this in the manuscript.

Reviewer #2 (Remarks to the Author):

Authors answers are fully satisfactory.

REVIEWERS' COMMENTS:

Reviewer #1 (Remarks to the Author):

Authors have partially addressed issues that I raised in the last round of review. Although the beta-catenin S37A rescue experiment is good, AXIN2, the most widely studied b-catenin target gene, is not affected by TNKS inhibitor or PRXII siRNA in SW480 cells. Down-regulation of several other genes in SW480 cells might not be mediated by b-catenin inhibition. Authors should minimally comment on this in the manuscript.

Answer: Since our GSE analysis using RNA-Seq data indicated down-regulation of total 384 genes by PrxII siRNA in SW480 cells, we acknowledge that down-regulation of some genes might not be mediated by β -catenin inhibition. However, we pointed that at least thirteen and eleven genes down-regulated by PrxII siRNA and TNKS inhibitor, respectively, in SW480 cells were obtained based on the public list of β -catenin target genes from the Wnt homepage (<http://web.stanford.edu/group/nusselab/cgi-bin/wnt/>) and the previous *BMC Genomics* paper (PMID: 24467841). Although we cannot explain the mechanism underlying that AXIN2 is affected by PrxII siRNA in HT29 cells but not in SW480 cells, we thought that several other β -catenin target genes, which were down-regulated by PrxII siRNA only in SW480 cells, might be mediated by a distinct β -catenin-associated transcription complex as hinted from the previous paper (PMID: 24419084). This comment is now added to the text (page 8, 3rd line).

Reviewer #2 (Remarks to the Author):

Authors answers are fully satisfactory.